# ONE MORE GLANCE WITH SHARP EYES: RETHINKING LIGHTWEIGHT CAPTIONING AS A PRACTICAL VISUAL SPECIALIST

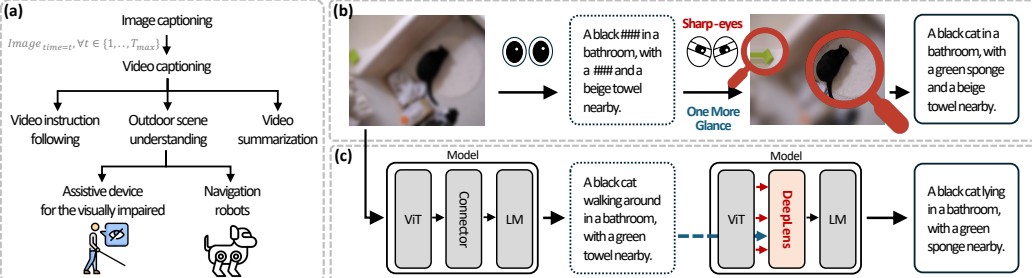

Figure 1: **(a)** Image captioning serves as the cornerstone of diverse applications, such as assistive navigation systems. However, existing captioning models are often too computationally heavy for deployment or suffer from limited performance. **(b)** The natural process of image description is illustrated: humans first take in the overall scene, then glance at specific regions to notice finer details. **(c)** Our Sharp-Eyed Refinement framework mimics the human tendency to take a second, more focused glance, allowing the captioning specialist to revise and improve its captioning outputs.

## ABSTRACT

Image captioning is fundamental for applications like video-grounded chatbot systems and navigation robots, yet deploying such models on local devices is challenging due to the high computational demands of multimodal LLMs (MLLMs). To address this, we first build lightweight captioning models using a 125M-parameter language model, 56 times smaller than LLaMA-7B, and evaluate their performance not only on single-sentence but on detailed captioning tasks. Surprisingly, we find that our model can achieve performance comparable to MLLMs, suggesting its potential to serve as a strong captioning specialist for on-device applications. While promising, our model also exhibits a limitation: like other MLLMs, it suffers from occasional captioning errors. We investigate the underlying causes and observe that the problems stem from ineffective attention mechanisms and limited visual representations. To alleviate them, we develop a novel captioning framework, Sharp-Eyed Refinement, which enhances caption quality by refining coarse descriptions into more precise captions. At its core, DeepLens improves visual grounding by re-examining the informative regions identified in the initial glance. Experimental results demonstrate the superiority of our model over both recent lightweight captioning methods and MLLMs in detailed captioning and even in long-range video QA tasks. We provide a summary of our work in Section A.

## 1 INTRODUCTION

Recent progress in image captioning has been driven by the emergence of MLLMs [44, 38], which exploit the advanced language processing capabilities of LLMs [95]. Building on these advances, image captioning has become a crucial component in various applications, as shown in Figure 1 (a). For example, video-based chatbot systems utilize frame-wise *caption* generation for temporal understanding [76, 84, 78], while navigation robots construct graph-structured scene *descriptions* to operate in complex environments [83, 28].

Despite recent progress, deploying such technologies on local devices remains a significant challenge due to the large computational demands of MLLMs [29]. Motivated by this, we explore lightweight captioning by implementing a captioning model using OPT-125M [93], a model 56 times smaller than LLaMA-7B [67] in the LLaVA framework [44]. To assess its effectiveness, we evaluate it on both single-sentence and detailed captioning tasks, and find that our lightweight model achieves impressive performance. In fact, it not only surpasses previous lightweight captioning models [59,25,31] but also achieves performance comparable to MLLMs, such as LLaVA-7B [44] and Instruct-BLIP [15]. Taken together, these findings reveal that the complex reasoning capabilities of LLMs are less critical for image captioning tasks that primarily involve enumerating *factual* details.

Despite these remarkable results, our lightweight model is not without limitations. It occasionally generates inaccurate captions, an issue widely studied in MLLMs, such as LLM's hallucination [5,2] and the blindness of vision encoders [65, 66]. Since the majority of our model's parameters are dedicated to the visual encoder, we focus on visual blindness as a potential root cause. This issue means that since the vision encoder manipulates visual representation while perceiving the image with its eyes wide shut [66], the LLM often struggles to generate accurate outputs.

We take a step towards analyzing the limitations of current captioning models. First, we inspect the model's attention during a single pass captioning and find that the model struggles to focus on critical regions of the image, instead spreading its attention across the entire image. Moreover, visual representations from a vision encoder like CLIP [57] lack sufficient detail for accurate caption generation. Specifically, when reconstructing images using these features, significant deviations from the original images are observed. These results highlight the need for more focused and detailed visual representations to provide the LM with clear detection information.

To address these limitations, we propose a novel captioning framework, Sharp-Eyed Refinement, as illustrated in Figure 1. Our method mirrors the human process of image description, where an initial broad understanding of the image is followed by a more focused and detailed analysis. In our framework, our model generates a rough caption, then refines its description. At the heart of our framework is our multimodal connector, DeepLens, which facilitates efficient re-examination of both the image and the initial caption during refinement. The effectiveness of our framework is demonstrated through experiments on single-sentence, detailed captioning, and long-range video QA; representative results are summarized in Figure 2.

Our contributions are summarized as follows: (*i*) We investigate the practically important yet under-explored topic of lightweight captioning (Section 2). We study lightweight models and demonstrate their potential as strong captioning baselines (Section 3). (*ii*) Inspired by the human mechanism of 'one more glance with sharp eyes,' we introduce a novel refinement framework (Section 4, 5). (*iii*) We conduct extensive experiments to validate the effectiveness and efficiency of our proposed framework (Section 6). We hope these efforts will inspire further development of efficient captioning models.

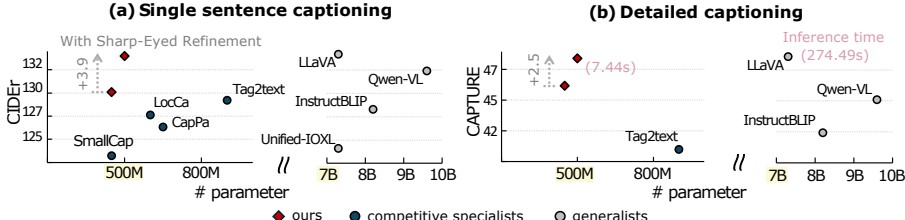

Figure 2: We evaluate our lightweight captioner on both single-sentence and detailed captioning tasks. It uses 93% fewer parameters and offers 97% faster inference than LLaVA-1.5-7B.

## 2 MOTIVATION & SCOPE

**Image Captioning as Foundational Technology.** Image captioning is the process of transforming visual information from an image into a natural language description [52, 24]. More than just a standalone task, this technology serves as a fundamental component for a wide range of applications, as outlined in Figure 1 (a). To process video-grounded chatbot systems [62, 77, 96, 84], image captioning is employed to generate descriptions of multiple frames, which are then integrated into LLMs as prompts to facilitate instruction following. Similarly, exploration robots [83,28] designed for disaster environments, as well as assistive robots for the visually impaired, utilize captioning to

encode observed scenes into a structured graph, supporting navigation and user interactions. In this work, we focus on image captioning as a crucial enabler for these applications.

**Real-World Deployment Challenges.** Recent studies aiming to implement such applications commonly utilize open-source MLLMs [76] or cloud-based APIs (*e.g.*, OpenAI API) [83] as image captioners. From a practical perspective, however, these approaches present significant challenges, as (1) open-source MLLMs require substantial computational resources that exceed the capabilities of edge devices like smartphones [7, 64], while (2) cloud-based APIs depend on a stable internet connection, which is likely to be unavailable in disaster situations. Table 1 presents the number of parameters used by recent MLLMs and their NVIDIA GPU memory requirements, assuming FP16 precision. Given that these applications require repetitive captioning process across multiple scenes [91], the feasibility of current models for real-world deployment remains uncertain. This resource-demand mismatch highlights the need for lightweight captioning models.

| Edge devices | | LLaVA-1.5 7B | LLaVA-NeXT 34B | LLaVA-OA 72B |
|---|---|---|---|---|
| iPhone 16 | Galaxy S25 | mPLUG-Owl3 8B | InternVL 40B | Qwen2-VL 72B |
| 8GB | 12GB | 16GB + | 68GB + | 140GB + |

Table 1: Available memory on edge devices and **GPU usage of recent MLLMs** by parameter size.

**Generalist vs. Specialist.** Following previous works [87, 63, 80], we use the term generalists to denote models trained on diverse data for broad purposes (e.g., current MLLMs [43, 37]), and specialists to denote models trained solely on task-specific data (e.g., small captioning models [59, 31])."

## 3 EXPLORING LIGHTWEIGHT CAPTIONING

To address the challenges outlined above, this section presents our design of a lightweight captioning specialist and provides comprehensive analyses of its effectiveness.

**Model construction.** To construct a lightweight captioning model, we initially focus on minimizing reliance on LLMs, as they contribute to the majority of computational overhead in MLLMs (*e.g.*, 96% in LLaVA-7B [44] is attributed to LLaMA [67]). To this end, we build our captioning specialist by replacing the LLaMA-7B used in LLaVA construction with OPT-125M [93], a language model 56 times smaller. It is notable that OPT-125M is significantly compact unlike modern small-scale language models exceeding 500M parameters.

**Implementation details.** We use the publicly available LLaVA-1.5 [43] code. Other than this modification, all training strategies follow the original code, including batch size, learning rate, and updating both the language model and the connector during fine-tuning while freezing the vision encoder. More details are available in Section H.1 and our source code[1]. All our experiments are conducted using two NVIDIA A6000 GPUs.

**Datasets and Evaluations.** The multimodal connector is first pretrained on the Caption Concept-balanced 558K [43], after which the model is fine-tuned on task-specific datasets such as COCO [14], DCI [70], or ShareGPT4V [11], as introduced later. We consider traditional metrics such as BLEU [56], CIDEr [71], and BERTScore [94], as well as MLLM-as-a-Judge [9, 8] with GPT-4o-mini.

### 3.1 RESULTS ON SINGLE SENTENCE CAPTIONING

MS COCO Captions [14], the most widely used captioning benchmark, consists of 113k images and 5 captions per image. One caption typically contains a single sentence of about 10 words. The results in Table 2 show that our model outperforms previous small captioning works [25, 72] that utilize fewer than 1B parameters. Specifically, our model achieves CIDEr scores that are 6.9 higher than SmallCap [59], which also uses OPT-125M as its LM backbone. Considering that our model does not involve any *newly proposed* methods, these results show surprisingly strong performance. We further discuss these results in Section D.1. Furthermore, we compare our model against MLLM generalists [13, 3]. Despite requiring significantly fewer computational resources, our model achieves comparable performance to these larger counterparts.

### 3.2 RESULTS ON DETAILED CAPTIONING

For this experiment, we fine-tune our model on a combination of the DCI [70] and ShareGPT4V [11] datasets, and additionally utilize the GLaMM [61] dataset. We then evaluate the model on the test splits of these datasets. Further details are provided in Section F.2.

---

[1] https://github.com/IclrAnonySubmit/ICLR26-submission-3588

| | model | venue | # data | # params | B@4 [56] | MET [16] | CIDEr [71] | BERT [94] | CLAIR [8] | GPT [9] |
|---|---|---|---|---|---|---|---|---|---|---|
| | MS COCO [14] | | | | | | | | | |
| G | InstructBLIP [15] | NeurIPS23 | 130M> | 8.2B | 38.0 | 29.4 | 127.8 | 69.1 | - | - |
| | Unified-IOXL [47] | ICLR23 | 130M | 7.3B | 37.0 | 29.5 | 123.6 | 68.2 | - | - |
| | Shikra [10] | arXiv23 | - | 7.2B | - | - | 117.5 | - | - | - |
| | Qwen-VL [3] | arXiv23 | 1.5B | 9.6B | 39.1 | 30.1 | 131.9 | **69.8** | 77.8±3.4 | 2.89±0.11 |
| | LLaVA-1.5 [43] | CVPR24 | 1M | 7.3B | 39.4 | 29.5 | 133.7 | 69.4 | 78.1±3.8 | 2.93±0.10 |
| | Cambrian [65] | NeurIPS24 | 2M | 10.5B | **40.1** | **30.9** | **137.5** | - | **78.2±3.2** | **3.02±0.13** |
| S | I-Tuning [50] | ICASSP23 | 0.5M | 250M | 35.5 | 28.8 | 120.0 | - | - | 2.50±0.10 |
| | CapPa [69] | NeurIPS23 | 1B | 650M | - | - | 125.8 | - | - | 2.67±0.08 |
| | LocCa [72] | NeurIPS24 | 1B | 600M | - | - | 127.1 | - | - | 2.66±0.11 |
| | SmallCap [59] | CVPR23 | 0.5M | 450M | 37.6 | 28.7 | 122.7 | 67.2 | 73.7±3.9 | 2.46±0.10 |
| | Tag2Text [25] | ICLR24 | 4M | 900M | 38.4 | 30.0 | 128.7 | 69.3 | 76.1±3.1 | **2.78±0.08** |
| | ViPCap [31] | AAAI25 | 4M | 225M | 37.7 | 28.6 | 122.9 | - | - | - |
| | Ours | | 1M | 450M | **39.4** | **30.3** | **129.6** | **69.4** | **76.3±3.0** | 2.74±0.06 |

| | model | venue | # data | # params | B@4 [56] | CIDEr [71] | BERT [94] | CAPT [18] | CLAIR [8] | GPT [9] |
|---|---|---|---|---|---|---|---|---|---|---|
| | ShareGPT4V [11] & DCI [70] | | | | | | | | | |
| G | Qwen-VL [3] | arXiv23 | 1.5B | 9.6B | 10.8 | 35.6 | 37.2 | 48.4 | 57.5±3.2 | **3.05±0.08** |
| | LLaVA-1.5 [43] | CVPR24 | 1M | 7.3B | 10.6 | 36.1 | 36.6 | 48.6 | ± | ± |
| | EyesWideShut [66] | CVPR24 | 1M | 7.6B | 11.6 | 36.5 | 37.2 | 49.0 | - | - |
| | Cambrian [65] | NeurIPS24 | 2M | 10.5B | **13.3** | **38.7** | **38.1** | 50.1 | **58.2±3.1** | 3.03±0.09 |
| S | SmallCap* [59] | CVPR23 | 0.5M | 450M | 14.5 | 20.1 | 28.9 | 23.3 | - | - |
| | Tag2Text* [25] | ICLR24 | 4M | 900M | 17.8 | 32.5 | 40.7 | 40.1 | 54.2±3.1 | 2.72±0.14 |
| | Ours | | 1M | 450M | **18.0** | **40.5** | **43.1** | **45.9** | **54.6±3.4** | **2.74±0.12** |

| | model | venue | # data | # params | B@4 [56] | CIDEr [71] | BERT [94] | CAPT [18] | CLAIR [8] | GPT [9] |
|---|---|---|---|---|---|---|---|---|---|---|
| | GLaMM [61] | | | | | | | | | |
| G | LLaVA-1.5 [43] | CVPR24 | 1M | 7.3B | 8.8 | 23.4 | 35.1 | 40.0 | 53.8±4.0 | 3.02±0.10 |
| S | Ours | | 1M | 450M | **16.5** | **29.1** | **38.7** | **42.0** | 51.8±4.1 | 2.64±0.09 |

Table 2: **Comparison of captioning performance.** We evaluate our specialist and existing captioning models. Despite not introducing any newly proposed methods, our model achieves strong performance. Here, 'G' represents a MLLM generalist, while 'S' denotes the small captioning model. '*' indicates models that are fine-tuned in this study, as they were not trained for detailed captioning tasks.

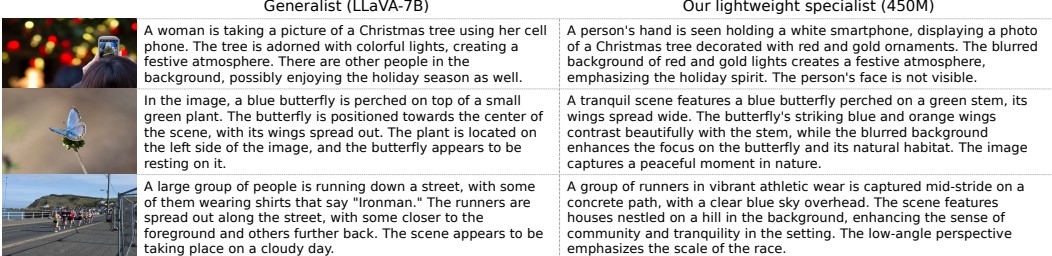

| Generalist (LLaVA-7B) | Our lightweight specialist (450M) |
|---|---|
| A woman is taking a picture of a Christmas tree using her cell phone. The tree is adorned with colorful lights, creating a festive atmosphere. There are other people in the background, possibly enjoying the holiday season as well. | A person's hand is seen holding a white smartphone, displaying a photo of a Christmas tree decorated with red and gold ornaments. The blurred background of red and gold lights creates a festive atmosphere, emphasizing the holiday spirit. The person's face is not visible. |
| In the image, a blue butterfly is perched on top of a small green plant. The butterfly is positioned towards the center of the scene, with its wings spread out. The plant is located on the left side of the image, and the butterfly appears to be resting on it. | A tranquil scene features a blue butterfly perched on a green stem, its wings spread wide. The butterfly's striking blue and orange wings contrast beautifully with the stem, while the blurred background enhances the focus on the butterfly and its natural habitat. The image captures a peaceful moment in nature. |
| A large group of people is running down a street, with some of them wearing shirts that say "Ironman." The runners are spread out along the street, with some closer to the foreground and others further back. The scene appears to be taking place on a cloudy day. | A group of runners in vibrant athletic wear is captured mid-stride on a concrete path, with a clear blue sky overhead. The scene features houses nestled on a hill in the background, enhancing the sense of community and tranquility in the setting. The low-angle perspective emphasizes the scale of the race. |

Figure 3: **Qualitative results on detailed captioning.** Despite its small size, our specialist generates effective image captions, with additional examples shown in Figure 15. For comparison, results from generalist models are provided in Figure 12 and Figure 13.

We initially anticipated that our specialist would struggle with this challenging task, given that OPT-125M possesses limited language processing capabilities, presumably due to its small parameter count, as shown in Table 4. However, the results in Table 2 and Figure 3 reveal that our model performs surprisingly well on detailed captioning tasks, contrary to our expectations. It should be noted that the vision encoder remains frozen while only OPT-125M is updated, meaning that detailed caption generation is largely handled by OPT-125M, with the vision encoder offering visual cues.

**Results Summary.** Although MLLMs rely on the complex reasoning abilities of LLMs for tasks such as VQA and instruction following, our experiments indicate that captioning—primarily involving the enumeration of factual objects—does not significantly require such capabilities. This observation suggests that a lightweight captioning model can serve as an efficient and practical alternative for captioning-based applications.

| Input prompt | Response | Input prompt | Response |
|---|---|---|---|
| What are we having for dinner? | A nice dinner with a friend. I'm going to have to go to bed. I'm going to <EOS> | A Italian street scene features | a man and a woman in a car, with a man in a car, with a woman in a car<EOS> |
| What should we eat for a snack? | We eat for snack. We eat for snack. <EOS> | A garden scene centers on | a garden of a garden of a garden <EOS> |

Figure 4: **Generation examples from OPT-125M.** Surprisingly, despite its limited capacity, our lightweight specialist performs image captioning effectively.

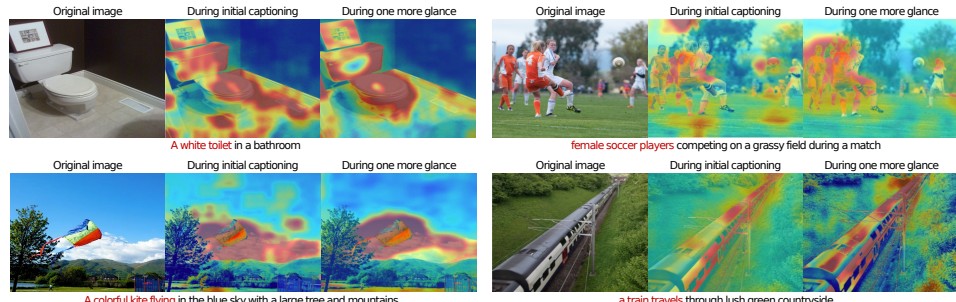

Figure 5: **Visualization of attention maps for highlighted words.** The left figure shows attention during single-pass caption generation, while the right figure illustrates the refinement process with one more glance guided by the initial caption.

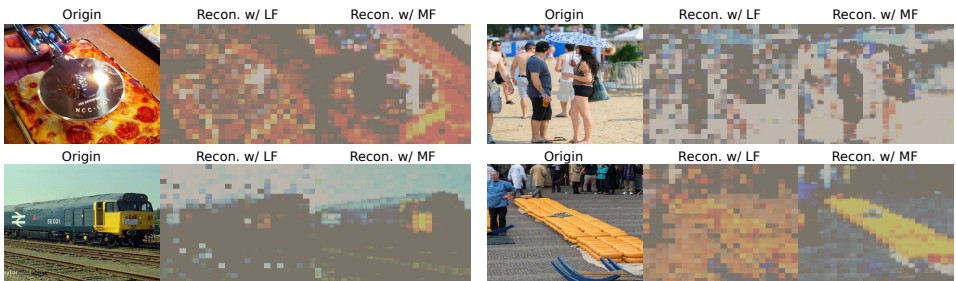

Figure 6: **Image reconstruction using visual representations** from CLIP [57]. Results are shown for last-layer features (*i.e.*, LF) and multi-layer features (*i.e.*, MF), where MF corresponds to representations extracted from layers 13, 18, and 23 of the 24-layer OpenAI CLIP ViT-L/14@336.

> *Note 1:* Factual image captioning does not significantly require the complex reasoning abilities of LLMs and can be handled by a lightweight language model.

Despite its impressive performance, our model shares similar limitations with other MLLMs [65, 46], occasionally producing incorrect predictions. Prior works have interpreted these errors from two perspectives: (*i*) the hallucination hypothesis, where LLMs, during response generation, describe non-existent elements in the image [5, 40]; (*ii*) the visual blindness hypothesis, where the visual encoder generates visual representations which are unclear or ambiguous [66]. Given that our model primarily consists of parameters dedicated to the vision encoder (e.g., 67% of total parameters), we hypothesize that the observed errors are more likely a result of visual blindness. In the following section, we provide an in-depth analysis of this issue and introduce our proposed solution.

## 4 WHY SHARP-EYED REFINEMENT?

In this section, we conduct a deeper investigation into our captioning specialist, formulating two hypotheses and validating them through toy experiments and visualizations. We then present our captioning framework to address these issues.

**Analysis 1.** (*Hypothesis.*) We propose a first hypothesis: they fail to focus on critical regions (*e.g.*, entities, attributes, relations), instead maintaining a blurry view of the entire image. One possible cause lies in the current captioning framework, which forces the model to process the entire image and produce a caption in a *single* pass. This is analogous to asking a human to describe an image in one glance, without attending to specific details. (*Setup.*) To investigate this issue, we analyze where in the image the model attends during the generation of specific words. The attention maps are visualized using code adapted from API [89], with further implementation details provided in Section H.3. (*Result.*) As illustrated in Figure 5, the model often exhibits diffuse attention and fails to focus on specific regions. These observations motivate the development of approaches that guide the captioning process toward more effective attention mechanisms.

> *Note 2:* Single-pass captioning can lead models to adopt broad attention, neglecting critical visual regions necessary for precise description.

$i_k$: image    $c_k$: ground-truth caption    $\hat{c}_k$: pseudo initial caption    *SFT*: supervised finetuning    $o_k$: caption output

Figure 7: **Inference and training pipeline for SeR.** The model is trained in two fine-tuning stages: (1) initial caption generation and (2) caption refinement. At inference, both an initial caption and a refined caption are produced, as shown by the solid and dashed lines in the rightmost figure.

**Analysis 2.** (*Hypothesis.*) Eyes Wide Shut [66] demonstrates that MLLMs suffer from visual deficiencies, as vision encoders such as CLIP [57] provide ambiguous representations that weaken performance. Based on this, we hypothesize that the widely used CLIP features may not provide sufficiently informative representations for generating accurate captions. (*Setup.*) Unlike prior studies, we examine this issue by reconstructing images from their visual features. The quality of the reconstructed images indicates how much visual detail the features preserve and, consequently, their effectiveness in supporting captioning tasks. For this experiment, we directly employ the decoder architecture from Masked Autoencoder (MAE) [22]. We keep the CLIP encoder frozen while training only the MAE decoder on the MS-COCO images. (*Result.*) As illustrated in Figure 6, the images reconstructed from CLIP embeddings exhibit significant deviations from the original images and appear less clear. This result indicates that such visual representations may not provide sufficiently detailed visual information for captioning.

> *Note 3:* CLIP's final-layer features provide only coarse-grained visual information, limiting their suitability for fine-grained captioning tasks.

**Our proposed solution.** Building on the above insights, we propose a novel captioning framework, *Sharp-Eyed Refinement* (SeR), illustrated in Figure 1 and Figure 7. The framework is inspired by the human description process, where one first takes an initial glance and then sharpens the eyes to capture finer details. Concretely, the model generates an initial caption that is subsequently refined by the LM through our *DeepLens*. This refinement process resembles approaches in LLM research such as self-refinement [51,33,55], where models improve their outputs by revisiting earlier generations. By extending this idea to the multimodal domain, we uniquely integrate visual features into the refinement step, and, to the best of our knowledge, this constitutes the first work in this direction. Further implementation details are provided in the following section.

## 5    INSIDE OUR FRAMEWORK: DEEPLENS AND LEARNING STRATEGY

In this section, we take a closer look at our framework and its effectiveness. At the core lies *DeepLens*, a multimodal connector that, unlike prior designs [44,38,3], leverages two new inputs: (*i*) previously generated captions and (*ii*) multi-layer features from the vision encoder. These inputs guide our model to attend to salient regions, enrich visual representations, and enable effective caption refinement.

**Look at what matters.** Imagine receiving an image along with the caption "A cat relaxing on a brown chair" and being instructed to refine it. In such a scenario, humans naturally focus on key elements like the cat and the chair. We attempt to employ a similar mechanism by providing the initially generated caption as an input for DeepLens. This design enables the module to identify and attend to the relevant visual regions. Similar to the experiment in Section 4, we also inspect the attention maps while the model takes a second glance using the previous captions, as shown in Figure 5. We observe that the model focuses more precisely on regions corresponding to specific words, indicating that it attempts to look at what matters in the image.

**Look in detail.** To mitigate the limitation of coarse-grained visual features, some studies [65,66,27] proposed using additional vision encoders, but this comes at the cost of additional model parameters. For example, Interleaved MoF [66] adds 300M parameters for DINOv2 [54]—a substantial 60% increase relative to our 500M-parameter model. Rather than introducing additional encoders, we focus on maximizing the utility of the existing vision encoder. To this end, we design DeepLens

| | Initially generated captions | After Sharp-eyed Refinement |
|---|---|---|
| | A striking blue and yellow train engine is stationed on a railway track, ready for its next journey. A red and white train car is visible in the background, set against a clear blue sky with a few clouds. The scene captures the essence of railway travel. | A striking blue and yellow train engine is stationed on a railway track, ready for its next journey. A black and grey car is visible in the background, set against a clear blue sky with no visible clouds. The scene captures the essence of railway travel. |
| | A woman in a white t-shirt and blue jeans is feeding a light brown sheep in a rustic barn. The sheep, with white coats and brown spots, stand on straw, while a black bucket and a wooden fence frame the scene. The image captures a peaceful moment in rural life. | A woman in a white t-shirt and blue jeans is petting a cream-colored sheep in a rustic barn. The sheep, with thick woolly coats, stand or lie on straw, while a black bucket and a wooden fence frame the scene. The image captures a peaceful moment in rural life. |
| | A man in a white lab coat and black pants is standing in front of a line of orange cheese blocks, with a metal fence and people in the background. The cheese blocks have different shapes and sizes, and the man's face is blurred out. | A man in a blue coat and black pants is standing in front of a line of round orange cheese wheels, with a red rope barrier and people in the background. The cheese blocks have different shape and size, and the man's face is blurred out. |

Figure 8: **Examples of captions during SeR.** Initial outputs occasionally include inaccuracies, which are corrected through the refinement process. More results can be found in Section E.

to leverage multi-layer features, expecting them to provide semantically richer representations that enable more accurate captioning, inspired by FPN [42]. As demonstrated in the reconstruction experiment (Figure 6), utilizing these features yields clearer images. This suggests that the multi-level visual features indeed contain fine-grained information, thereby helping the LM recognize visual content in greater detail.

> *Note 4:* Encouraging attention to key regions via the initial caption, along with multi-level utilization of a single vision encoder, can be an effective and efficient strategy.

**Fine-tuning strategy.** Assume we have a training set $X = \{x_1, \ldots, x_N\}$, where each $x_k = (i_k, c_k)$ consists of an image $i_k$ and its corresponding ground-truth caption $c_k$; $N$ denotes the total number of such pairs. As illustrated in Figure 7, we adopt a two-stage fine-tuning strategy. In the first stage, we focus on enabling the model to generate an initial caption by following the training procedure of LLaVA. Specifically, we perform supervised fine-tuning on the MLP connector and the lightweight language model using this dataset.

In the second stage, we focus on teaching the model the refinement process. A naïve approach would be to directly use the momentary generated captions as initial inputs and the true captions $c_k$ as targets for refinement, but this can be misleading. Suppose our captioning specialist were to initially produce something like 'a brown table in front of a window' while the true caption is 'a cat sitting on a chair'; these two descriptions share little in common and provide no meaningful signal for progressive refinement. We believe that training on such disparate pairs would encourage the model to ignore the provided initial caption and generate a wholly new one, rather than learn to refine it. We provide additional explanations in Section D.5. To address this challenge, we construct pseudo-initial captions $\hat{c}_k$ that remain close to their corresponding ground-truth $c_k$. In practice, we leverage GPT-4o-mini [1] to slightly alter entities, attributes, or relations within $c_k$. For example, given the true caption 'a cat sitting on a chair', the pseudo-initial caption might become 'a dog sitting on a chair,' preserving the overall structure while introducing minor variations.

Leveraging these pseudo-initial captions, each pair $(i_k, \hat{c}_k)$ is fed into DeepLens together with the language model, and optimization is performed using a next-token prediction loss (*i.e.*, supervised fine-tuning). Our fine-tuning schedule consists of 10 epochs for the first stage and 2 epochs for the second stage. Further details of the training setup and hyperparameters are provided in Section H.1, and we describe the generation strategy for $\hat{c}_k$ in Section F.1, along with an ablation study on refinement data in Section D.5.

**Rationale for refinement fine-tuning.** Let $\pi_\theta$ denote the language model. Each pseudo-initial caption $\hat{c}_k$ deviates from the ground truth $c_k$ at only a few token positions $E_k = \{t \mid \hat{c}_{k,t} \neq c_{k,t}\}$. Under the sequence-level cross-entropy objective

$$\mathcal{L}(\theta) = -\mathbb{E}\Big[\sum_j \log \pi_\theta\big(c_{k,j} \mid i_k, \hat{c}_k, c_{k,<j}\big)\Big], \tag{1}$$

gradients are likely to be primarily concentrated on tokens in $E_k$, leading to a form of *targeted optimization*, in which the model retains the correct parts of $\hat{c}_k$ while rewriting only the erroneous ones. Writing $\Delta_k(\theta) = \log \pi_\theta(c_k \mid i_k, \hat{c}_k) - \log \pi_\theta(\hat{c}_k \mid i_k, \hat{c}_k)$, we have $\mathcal{L}(\theta) \propto -\mathbb{E}[\Delta_k]$; hence, minimizing $\mathcal{L}$ is equivalent to maximizing the expected margin $\Delta_k$, thereby directly increasing the likelihood of the refined caption relative to its *flawed* precursor. In effect, each gradient step encourages DeepLens to focus on visual regions likely responsible for errors in the initial caption and guides the language model to better interpret these refined features, resulting in more accurate

| MS COCO [14] | | | | | | | | | |
|---|---|---|---|---|---|---|---|---|---|
| model | #params | B@4 [56] | gain | CIDEr [71] | gain | CLAIR [8] | gain | GPT [9] | gain |
| LLaVA-1.5 [43] | 7.3B | 39.4 | - | 133.7 | - | 78.1±3.8 | - | 2.93±0.10 | - |
| Our specialist | 450M | 39.4 | - | 129.6 | - | 76.3±3.0 | - | 2.74±0.06 | - |
| + SeR with ①+② | 500M | 39.9 | +0.5 | 133.5 | +3.9 | 77.6±2.9 | +1.3 | 2.82±0.09 | +0.08 |
| + SeR with ① | 500M | 39.6 | +0.2 | 131.9 | +2.3 | - | - | - | - |
| + SeR with ② | 500M | 39.6 | +0.2 | 132.3 | +2.7 | - | - | - | - |
| Single glance with ② | 450M | 39.7 | +0.3 | 130.6 | +1.0 | - | - | - | - |

| ShareGPT4V [11] & DCI [70] | | | | | | | | | |
|---|---|---|---|---|---|---|---|---|---|
| model | #params | CIDEr [71] | gain | CAPT [18] | gain | CLAIR [8] | gain | GPT [9] | gain |
| Cambrian [65] | 10.5B | 38.7 | - | 50.1 | - | 58.2±3.1 | - | 3.00±0.10 | - |
| Our specialist | 450M | 40.5 | - | 45.9 | - | 54.6±3.4 | - | 2.74±0.12 | - |
| + SeR with ①+② | 500M | 43.6 | +3.1 | 48.4 | +2.5 | 57.7±3.0 | +3.1 | 3.02±0.12 | +0.28 |
| + SeR with ① | 500M | 42.8 | +2.3 | 47.1 | +1.2 | 55.8±3.1 | +1.2 | 2.78±0.11 | +0.04 |
| + SeR with ② | 500M | 43.1 | +2.6 | 47.6 | +1.7 | 56.8±3.4 | +2.2 | 2.88±0.12 | +0.14 |
| Single glance with ② | 450M | 42.5 | +2.0 | 46.5 | +0.6 | 56.3±2.9 | +1.7 | 2.90±0.11 | +0.16 |

| GLaMM [61] | | | | | | | | | |
|---|---|---|---|---|---|---|---|---|---|
| model | #params | CIDEr [71] | gain | CAPT [18] | gain | CLAIR [8] | gain | GPT [9] | gain |
| LLaVA-1.5 [43] | 7.3B | 23.4 | - | 40.0 | - | 53.8±4.0 | - | 3.02±0.10 | - |
| Our specialist | 450M | 29.1 | - | 42.0 | - | 51.8±4.1 | - | 2.64±0.09 | - |
| + SeR with ①+② | 500M | 30.4 | +1.3 | 42.8 | +0.8 | 53.4±3.8 | +1.6 | 2.88±0.11 | +0.24 |

Table 3: **Quantitative results of Sharp-Eyed Refinement (SeR).** The results demonstrate the effectiveness of SeR in improving caption quality. We also present an ablation study on the key inputs to DeepLens: ① initial captions and ② multi-layer features.

captions. We note that this targeted optimization resembles the philosophy of Direct Preference Optimization (DPO) [58], if we consider $c_k$ and $\hat{c}_k$ as preferred and less-preferred responses. In contrast to DPO, which treats both responses symmetrically during optimization, our method offers a new perspective by assigning them distinct roles as input and target.

**Inference strategy.** Given an image $i_k$ and instruction $p$, the model generates an initial caption $o_{initial}$, storing intermediate ViT [19] features in a buffer $B$ to avoid redundant computation. $o_{initial}$ is then fed into DeepLens and the LM to produce the refined output $o_{refined}$. Unless otherwise specified, we employ a single-step refinement. A discussion and experimental results for a repeated process are presented in Section D.6 and Section B.6, respectively.

## 6   EMPIRICAL VALIDATION

**Ablation Study of SeR.** The results in Table 3 and Figure 8 demonstrate the effectiveness of SeR in improving caption quality. We further conduct an ablation study on the key inputs to DeepLens—① initial captions and ② multi-layer visual features—to examine their individual contributions. The refinement process improves initial captions by +3.9 and +2.5 CIDEr points for single-sentence and detailed captioning tasks, respectively. Although integrating only multi-layer features (*i.e.*, using ②) yields modest gains, we observe that its effectiveness may be constrained by the single-pass generation mechanism. Moreover, our method introduces approximately 50M additional parameters for DeepLens and one extra inference step with OPT-125M. Nevertheless, as shown in Table 9, the computational overhead remains minimal relative to the substantial improvements achieved. We further validate this efficiency on a resource-constrained device, **Jetson Nano**, with results reported in Section B.3.

| model | time |
|---|---|
| LLaVA | 274.49 s |
| Ours | 5.55 s (97.97%↓) |
| + SeR | 7.44 s (97.28%↓) |

Figure 9: Inference time required to generate captions for 100 streaming images.

**Evaluation on Long Range Video Question Answering.** Since captioning models serve as foundational components in downstream applications as mentioned in Section 2, it is important to verify their utility in a practical task. To this end, we evaluate our model on the Long-Range VQA task introduced in [91]. Following the baseline setup of Zhang et al. [91], each captioner generates captions for multiple frames from videos spanning several minutes to hours. These captions are then fed into an LLM to answer video-related questions. As shown in Table 4, our model achieves competitive accuracy with significantly fewer parameters and faster inference than MLLM generalists, demonstrating its potential for real-world applications. The reported time includes both the captioning and LLM inference steps, assuming a 10-minute video processed at the frame level. The small specialists [59, 25], including ours, use weights fine-tuned on ShareGPT [11] & DCI [70].

> **Note 5:** Although lightweight captioners play an important role, they have received limited attention. Our proposed baseline and framework highlight their promise as efficient solutions for on-device applications.

| LLM | Captioner | LLaVA-1.0 [44] | BLIP-2 [38] | LLavA-1.5 [43] | SmallCap* [59] | Tag2text* [25] | Our specialist | + SeR |
|---|---|---|---|---|---|---|---|---|
| | #params | 7.3B | 7.4B | 7.3B | 450M | 900M | 450M | 500M |
| Qwen2.5 14B [85] | accuracy | 47.6 | 50.6 | **51.1** | 41.8 | 47.1 | 49.3 | **50.8** |
| | time | 29m 20s | 29m 44s | 29m 20s | 4m 45s | 7m 14s | 4m 53s | 5m 10s |

Table 4: **Evaluation on Long-Range VideoQA.** We follow the baseline setup of LVQA [91], replacing the captioner with our specialist or MLLM generalists. Captions from all frames are aggregated and fed into Qwen2.5-14B to perform the final video question answering. Higher-quality captions from our specialist contribute to competitive accuracy with significantly fewer parameters.

## 7 RELATED WORK

**Multimodal Large Language Models (MLLMs)** have attracted considerable research attention due to their versatile applications, such as chat-bots [81]. Early approaches integrated contrastive image-language pretrained models [74] with powerful LLMs, enabling complex reasoning. The development of instruction-based datasets [44] and innovative training strategies [36,87] have further accelerated progress, substantially improving MLLM performance and broadening their capabilities. Despite these achievements, most MLLMs heavily depend on large-parameter LLMs, making deployment on memory-constrained devices unfeasible. This limitation will likely restrict access for a significant portion of users worldwide.

**Image Captioning Models.** Recent advances in image captioning have improved training efficiency and descriptive fidelity. Approaches such as CaMEL [6] and SmallCap [59] emphasize minimizing *trainable* parameters by leveraging mean-teacher distillation and employing retrieval augmentation, while Tag2Text [25] and LoCCa [72] introduce novel mechanisms such as dedicated tagging and location-aware refinement to improve caption quality. Unlike existing approaches that focus on reducing *trainable* parameters, or rely on single-pass inference—potentially missing crucial details—our method prioritizes *inference* efficiency considering on-device operation and systematically addressing the limitations of single-pass generation.

**Visual Blindness in VLMs.** Despite significant advancements, MLLMs still face limitations in their visual capabilities, hindering their practical applications. Eyes Wide Shut [66] demonstrated that even GPT-4V [1] struggles with basic visual questions. Research on this topic typically points to two main sources of failure: one relates to the language decoder, which can hallucinate details not present in the image [5], while the other focuses on the visual encoder, which may provide ambiguous visual information. Several studies, including Cambrian [65], believe the visual encoder is a critical bottleneck. We also concentrate on the visual issue, particularly within the context of our lightweight model, where the visual encoder accounts for a significant portion of the parameters. Furthermore, we introduce a novel operational framework to improve visual grounding.

**Self-Refinement in LLMs.** Humans often refine their writing through iterative review to enhance clarity and precision [51,20]. Recent research has applied this refinement concept to LLMs, introducing techniques. For instance, Self-Refine [33,60,55] enabled models to autonomously critique and iteratively enhance their outputs. Unlike such approaches limited to the LLM domain, our method introduces refinement in a multimodal context, guided by both language and vision.

## 8 CONCLUSION & BROADER IMPACT

We presented lightweight captioning with Sharp-Eyed Refinement for efficient deployment of visual understanding systems on edge devices. Our specialist OPT-125M-based model achieves comparable performance to large multimodal models while significantly reducing parameters (93%↓) and inference time (97%↓). Specifically, we identified ineffective attention mechanisms and coarse-grained visual representations as key causes of visual blindness, and introduced Sharp-Eyed Refinement along with its core module, DeepLens. This framework focuses on salient visual regions guided by the initial caption and leverages finer-grained visual representations. Extensive experiments confirm strong performance, validating our approach as a practical solution for on-device applications.

**Broader Impact.** We demonstrate the potential of small-scale specialists, highlighting their surprising effectiveness. As a next step, future research may explore iterative refinement strategies, integrating external tools (e.g., zoom or crop, as in GPT-4o [53]), and designing a unified multimodal connector for both the initial and secondary glance. More research questions that aim to enhance both captioning performance and efficiency in real-world applications are provided in Section D.6.

REPRODUCIBILITY STATEMENT

We aim to make our results fully reproducible. The anonymized source code for our strong baseline can be found in the implementation described in Section 3. For model training, we primarily rely on publicly available datasets, as discussed in Section 3. In addition, the pseudo-initial caption data we created is released through a Google Drive link embedded in the same code repository. The prompts used for generating this data are provided in Section G, and most of the model training parameters are detailed in Section H.1. Implementation details are described in Section H.2, Section H.3, Section H.4, and Section H.5.

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

# Appendix

## Table of Contents

# A    SUMMARY OF OUR WORK

### ✅ Motivation

1. Image captioning serves as the cornerstone of diverse applications such as assistive devices.
2. However, running multimodal large language models for this task is challenging for edge devices.
3. We highlight the practical importance of lightweight captioning.

### ✅ What We Have Demonstrated?

1. (*Note1*) Factual image captioning does not significantly require the complex reasoning abilities of LLMs and can be handled by a lightweight language model.
2. (*Note2*) Single-pass captioning can lead specialists to adopt broad attention, neglecting critical visual regions necessary.
3. (*Note3*) CLIP's final-layer features provide only coarse-grained visual information, limiting their suitability for fine-grained captioning tasks.

### ✅ Novel Caption Refinement Framework

- **What is it?** Humans first take in the overall scene, then glance at specific regions to notice finer details. Our Sharp-Eyed Refinement framework mimics this human tendency, allowing the captioning specialist to revise and improve initial descriptions.
- (*Note4*) Encouraging attention to key regions via the initial caption, along with multi-level utilization of a single vision encoder, can be an effective and efficient strategy.
- **Why novel?** To the best of our knowledge, this is the first work to apply multimodal refinement that jointly utilizes visual features and the model's initial textual output.
  - *NLP community*: Humans often refine their writing, and coders revise their code through iterative review. Recent research has applied this refinement/correctness concept to LLMs [51, 60, 82, 32]. Unlike such approaches limited to the LLM domain, our method introduces refinement in a multimodal context.
  - *Multimodal LLMs community:* Existing models [44, 3, 97] heavily rely on the complex reasoning capabilities of LLMs and typically generate outputs in a single pass. We uniquely demonstrate the effectiveness of revisiting and refining its own textual outputs.
  - *Visual Blindness community:* Several studies [65, 66] believe the visual encoder is a critical bottleneck and adapting multiple vision encoders. However, this requires additional parameters. Since our research focuses on a lightweight model, we maximize the utility of the existing vision encoder through multi-level utilization.
  - *Lightweight captioning community:* This field has been gradually declining since the remarkable capabilities of LLMs were discovered. In this work, we revisit the practically important yet underexplored topic of lightweight captioning models.
    - ∗ Unlike ours, some methods [49, 59, 73] utilize prior captions obtained through a heavy image-text retrieval process. For example, SmallCap performs similarity matching between a given image and 500,000 candidate captions at each iteration.
    - ∗ Unlike ours, several works [25, 48] incorporate object detection or tagging procedures. However, despite these extra components, their captioning performance has been limited.
    - ∗ Unlike ours, which refines text output generated via a full forward pass, certain works [49, 12] adopt Diffusion Transformers and denoise latent text embeddings. Moreover, they provide little motivation or analysis as to why such a process is important from the perspective of utilizing visual cues more effectively.

### ✅ Broad Comparisons

1. This is the first paper to demonstrate the amazing capability of a lightweight captioning model beyond single-sentence captioning, including not only detailed captioning but also long-form VideoQA.
2. We compared not only existing small captioning models but also large multimodal models, demonstrating the potential of our model as a strong visual specialist.

3. Through these extensive experiments, (***Note5***) we show that our proposed baseline and framework have the potential of lightweight captioning for on-device applications.

✅ **Positive Influences**

1. **Industry:** Our model can be easily applied in industrial scenarios such as assistive devices for the visually impaired or navigation robots. Our model runs successfully on actual edge devices like Jetson Nano.
2. **Academy:** For researchers with limited access to large-scale GPUs who are unable to operate LLMs, our factual image captioning model offers a promising baseline.

---

# B Additional Experiments

## B.1 When GPT meets 'one more glance'

In this work, we suggest that allowing the model to take one more glance, guided by the initial caption and aligned visual features, offers an effective approach, as discussed in Note 4. We further explore how OpenAI's GPT models, which have demonstrated remarkable performance, might be influenced by our refinement framework. While our framework consists of two components, taking one more glance and refining with sharp eyes through DeepLens, we apply only the former as modifying their internal architecture is not feasible. The results are shown in Figure 10, where GPT-4-turbo is used. As a commercial model with a sufficiently large number of parameters, it often produces an accurate initial caption without noticeable errors, but we still observed occasional inaccuracies in its outputs. To explore whether refinement could help, we instructed the model to **take one more glance** and found that GPT can generate a refined caption by re-examining the given image. OpenAI's GPT-o3 reflects an awareness of this idea through the introduction of the Thinking with Images [53]. However, this concept remains less explored in the research community, suggesting the need for further investigation.

## B.2 When larger specialists meet 'one more glance with sharp eyes'

Building on our lightweight specialist, we naturally ask: since our framework and module offer a scalable methodology, can they also be effective when applied to larger multimodal models? To investigate this, we scaled the language model from OPT-125M to OPT-1.3B and also utilized LLaMA-2-7B. Specifically, we fine-tuned these models on the ShareGPT and DCI training datasets to serve as captioning specialists (*i.e.*, single pass). Next, we incorporated our Sharp-Eyed Refinement (*i.e.*, SeR) framework with DeepLens and further fine-tuned them using the datasets described in Section 5. As shown in Table 5, our framework consistently improves caption accuracy across different models. In particular, we observe gains of 1.2 and 0.9 points in CAPT [18] evaluation for OPT-1.3B and LLaMA-2-7B, respectively. Additionally, the results indicate that using larger language models to build captioning specialists leads to improved performance, observed in prior LLM scaling studies [30, 79, 24]. Despite the gains observed with larger models such as OPT-1.3B and LLaMA-2-7B, we note that their considerable size (roughly 10x and 56x larger than OPT-125M, respectively) may hinder real-world applicability. It is therefore crucial to continue exploring captioning systems that balance effectiveness with efficiency.

| ShareGPT4V [11] & DCI [70] | | | | | | | |
|---|---|---|---|---|---|---|---|
| Language model | vision encoder | LoRA [23] | total # params | CIDEr [71] single pass | CIDEr [71] + SeR | CAPT [18] single pass | CAPT [18] + SeR |
| OPT-1.3B [93] | CLIP ViT-L | × | 1.6B | 50.2 | **53.1** (+2.9) | 49.0 | **50.2** (+1.2) |
| LLaMA-2-7B [67] | CLIP ViT-L | ✓ | 7.3B | 57.3 | **61.7** (+4.4) | 52.7 | **53.6** (+0.9) |

Table 5: Captioning performance of larger specialists with and without Sharp-Eyed Refinement (SeR). SeR consistently improves accuracy across model scales, including OPT-1.3B and LLaMA-2-7B.

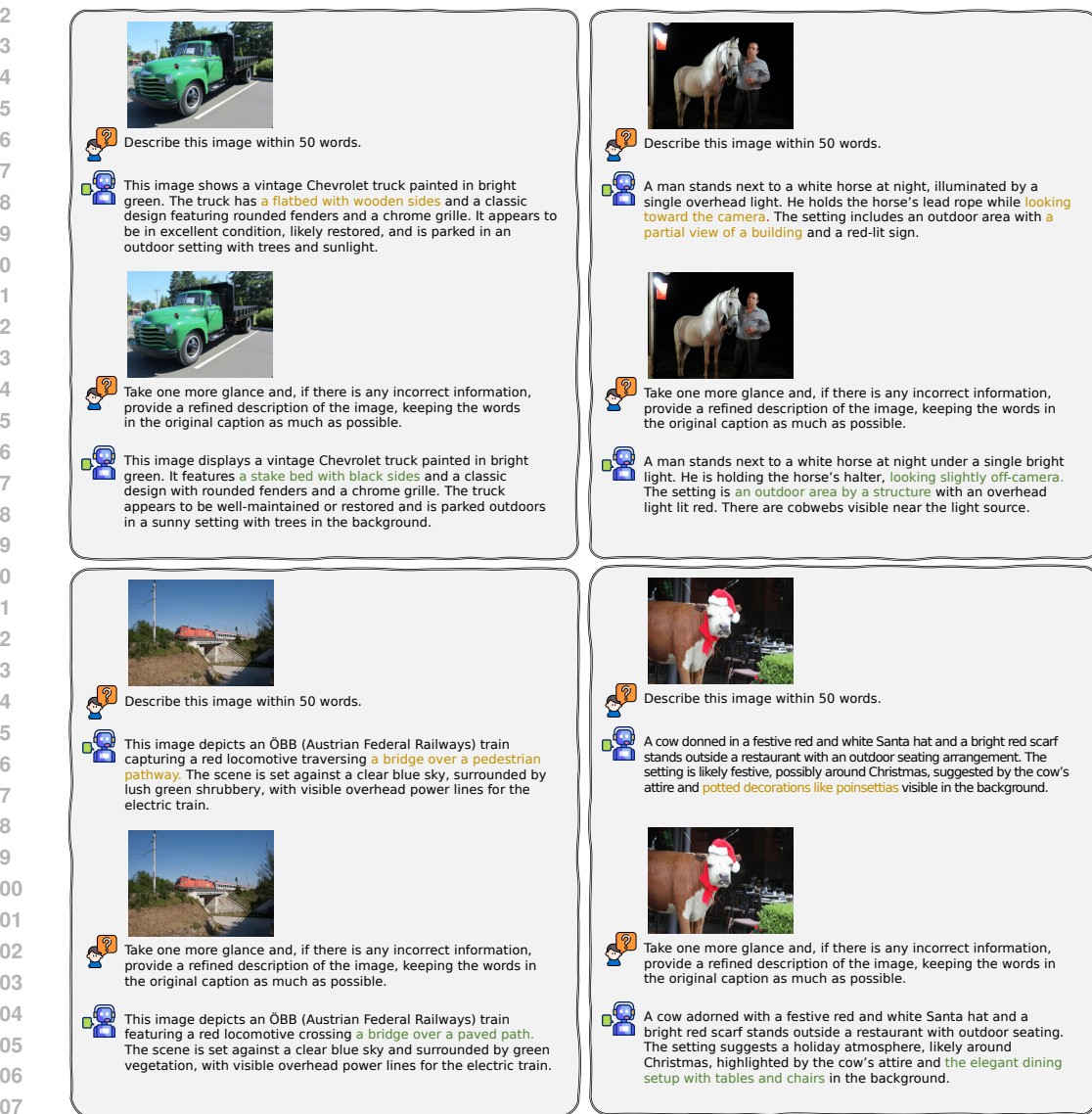

Figure 10: **When GPT meets 'one more glance'.** Example outputs from OpenAI's GPT before and after being prompted to take one more glance. The refined captions illustrate its potential responsiveness to visual re-examination.

### B.3 EVALUATION ON AN ACTUAL EDGE DEVICE

To highlight the practical value of our approach, we present an evaluation of our lightweight captioner on edge devices, where large-scale MLLMs are typically infeasible to deploy. Our core motivation is rooted in the observation that while MLLMs are powerful, they are computationally intensive and difficult to run in resource-constrained environments. In contrast, lightweight captioners remain underexplored despite their potential for real-world applications.

Here, we demonstrate that lightweight captioners can be effectively deployed on edge devices such as RTX 3090 and **Jetson Nano**. We evaluate captioning performance on MS COCO and ShareGPT4V&DCI datasets, and additionally measure inference time, memory usage, and power consumption. All models were executed for 100 iterations with a batch size of 1. The results in Table 6 show that our method consistently performs well across devices, and notably, remains fully operational on setups where models like LLaVA-1.5 fail to run. These findings support the

| Resource (RAM) | Data | Model | Mem. | Inf. time | Power. | B@4 [56] | CIDEr [71] | CLAIR [8] | GPT [9] |
|---|---|---|---|---|---|---|---|---|---|
| Jetson Nano (4G) | All | LLaVA-1.5-7B | **out-of-memory** | N/A | N/A | N/A | N/A | N/A | N/A |
| RTX 3090 (24G) | MS COCO [14] | Ours-500M | 3.2G | 5s | 230 W | 39.5 | 133.8 | 78.6±2.9 | 2.83±0.06 |
| Jetson Nano (4G) | | | 2.6G | 20s | 13 W | 39.5 | 133.8 | 78.6±2.9 | 2.73±0.09 |
| RTX 3090 (24G) | ShareGPT4V [11] & DCI [70] | Ours-500M | 3.2G | 5s | 230 W | 22 | 43.2 | 57.9±3.0 | 3.01±0.10 |
| Jetson Nano (4G) | | | 2.7G | 21s | 13 W | 22.2 | 42.9 | 57.4±2.9 | 3.02±0.11 |

Table 6: Evaluation results across different hardware resources and datasets.

| | ShareGPT4V [11] & DCI [70] | | | | | | |
|---|---|---|---|---|---|---|---|
| Metric | **origin** | +SeR | gain | **more data** | **distillation** | +SeR | gain |
| CIDEr [71] | 40.5 | 43.6 | +3.1 | 36.8 | 42.6 | 43.6 | +2.0 |
| CAPT [18] | 45.9 | 48.4 | +2.5 | 43.2 | 46.5 | 47.4 | +0.9 |

Table 7: Performance comparison of our lightweight captioner under different learning strategies: origin, more data, and distillation. Distillation from a strong teacher, especially when combined with Sharp-Eyed Refinement (SeR), leads to the best results.

deployability of our framework in edge scenarios and encourage further exploration of lightweight captioning for real-world assistive technologies.

### B.4 COMPARISON ON DIFFERENT LEARNING STRATEGIES

As writers frequently revise their drafts to improve clarity or tone, and programmers refactor code to enhance structure, we demonstrate that allowing multimodal models to refine their own outputs through our framework can be an effective strategy. In this subsection, we examine how our lightweight specialist performs under different learning strategies. The first strategy involves training with as much data as possible. Here, we train the model on a combined set of COCO, ShareGPT, DCI, and GLaMM to maximize data exposure (*i.e.*, more data). The second strategy explores distillation using teacher-generated captions. Specifically, we train our model on captions produced by LLaVA-1.6-34B [45] (*i.e.*, distillation). Our original strategy was straightforwardly fine-tuning the model exclusively on task-specific data (*i.e.*, origin). The results are summarized in Table B.4. For the 'more data' strategy, we observe limited performance, likely due to reduced task alignment caused by mixing heterogeneous datasets. In contrast, the 'distillation' strategy demonstrates that learning from a strong teacher can improve performance even in a single-pass setting. Notably, when combined with our refinement framework (SeR), it yields further gains, suggesting that distillation and SeR can be effectively integrated. Additionally, as noted in Section 5, the datasets we used share similarities with those in Direct Preference Optimization (DPO). Investigating the impact of training our model using reinforcement learning methods such as DPO could be a promising direction for future work.

### B.5 EFFICACY OF MULTI-LEVEL FEATURES WITH OTHER VISION ENCODERS

We investigate whether our strategy of selecting multi-level features can generalize to vision encoders beyond CLIP [57], which is our original setup. As shown in Table 8, the approach consistently improves performance across multiple encoders. Specifically, both SigLIPv2 [68] and DINOv2 [54] exhibit performance gains when equipped with multi-level features, demonstrating that our method is not limited to CLIP-based models. Interestingly, CLIP with multi-level features outperforms the combination of CLIP and DINOv2 in several evaluation metrics, while maintaining greater parameter efficiency. On the other hand, DINOv2 alone yields relatively lower performance, likely due to weaker alignment with language features. For these experiments, we pair OPT-125M with each vision encoder and fine-tune the resulting models on the ShareGPT [11]&DCI [70] datasets. Subsequently, we apply our Sharp-Eyed Refinement framework using the selected multi-level features and perform second fine-tuning with the datasets described in Section 5 of the main paper.

### B.6 ITERATIVE SHARP-EYED REFINEMENT

To explore the potential of iterative refinement beyond a single pass, we conduct additional experiments using captioning specialists with larger model capacity. This decision is motivated by our initial hypothesis that the constrained capacity of small models may have hindered the effectiveness of multi-step refinement. As shown in Table 9, applying multiple refinement iterations to the smallest language model, OPT-125M, results in only limited improvements. In contrast, we observe that multi-step refinement leads to performance gains when used with higher-capacity captioners. These

| Vision encoder | #params | indices of selected layers | CIDEr [71] | CLAIR [8] | GPT [9] |
|---|---|---|---|---|---|
| CLIP [57] | 300M | {23} | 42.8 | 55.8 | 2.78 |
| | 300M | {13, 18, 23} | 43.3 | 57.7 | 3.02 |
| CLIP [57]+DINOv2 [54] | 600M | {23} + {23} | 42.9 | 57.3 | 3.02 |
| SigLIPv2 [68] | 300M | {23} | 43.0 | 56.2 | 2.88 |
| | 300M | {15, 23} | **45.9** | 57.7 | 3.05 |
| | 300M | {13, 18, 23} | 45.5 | **58.2** | **3.07** |
| DINOv2 [54] | 300M | {23} | 32.8 | 48.6 | 2.55 |
| | 300M | {13, 18, 23} | 33.0 | 50.1 | 2.66 |

Table 8: For the captioning specialist, we evaluate different vision encoders and the corresponding selected layer indices.

| Language model | vision encoder | Benchmarks | Eval. matrix | initial cap. | refinement | refinement *2 | refinement *3 | refinement *4 |
|---|---|---|---|---|---|---|---|---|
| **OPT-125M** | | | | 129.6 | 133.5 | **133.8** | 133.1 | 133.4 |
| OPT-1.3B | CLIP ViT-L | MS COCO | CIDEr [71] | 133.7 | 135.7 | **138.1** | 138.1 | 138.5 |
| LLaMA-2-7B | | | | 137.4 | 141.4 | 141.6 | **141.9** | 141.7 |
| **OPT-125M** | CLIP ViT-L | ShareGPT4V | GPT [9] | 2.74±0.12 | **3.02±0.12** | 3.01±0.10 | 3.03±0.10 | 3.00±0.12 |
| OPT-1.3B | | &DCI | | 3.04±0.10 | 3.14±0.11 | 3.16±0.10 | **3.20±0.10** | 3.19±0.10 |

Table 9: Multiple iterative refinements with larger language models. This experiment demonstrates better refinement capability as model capacity increases.

findings underscore the potential of scaling up refinement strategies, as similarly observed in recent LLM-based studies [51,60,82,32]. We plan to further investigate this direction in future work.

For implementation, we construct larger captioning specialists based on two configurations: LLaMA-2-7B + CLIP-ViT-Large and OPT-1.3B + CLIP-ViT-Large. These models are initially fine-tuned on the ShareGPT and DCI training datasets via single-pass training. We then apply the Sharp-Eyed Refinement (SeR) framework with DeepLens in multiple iterations, followed by additional fine-tuning using the datasets described in Section 5.

# C DeepLens

## C.1 Architectural details

We have shown that our refinement framework can improve caption quality. At the core of it lies DeepLens, which we hypothesize plays a critical role by aligning multi-level visual features with semantic cues from the initial caption. As illustrated in Figure 11, DeepLens takes two new inputs: ① the previously generated caption, represented as token embeddings, and ② multi-layer features extracted from the vision encoder. Let $N$, $d$, and $m$ denote the number of visual tokens, the embedding dimension, and the number of selected layers, respectively. The selected features are concatenated along the channel dimension to retain hierarchical visual information. These fused inputs are then passed through Transformer layers following the BERT architecture [17]. The resulting output is provided to the language model, enabling refinement conditioned on a focused and semantically grounded visual representation.

## C.2 Ablation study

We conduct a series of ablation experiments to analyze the design choices behind DeepLens, as summarized in Table 10. The corresponding model variants are illustrated in Figure 11. Our study spans three key aspects: connector selection, refinement configurations, and the layer indexing strategy for visual feature extraction. **(I)** We begin by comparing different multimodal connectors used in the initial caption generation phase. While prior works have explored a range of designs, such as Cross-Attention [59,6], Q-Former [39], and Transformer-style modules, we find that the simple MLP connector from LLaVA [43] offers competitive performance. Notably, (b) and (f), which adopt multi-level features and a BERT-style Transformer block respectively, perform slightly better than the MLP-based connector. However, given the marginal gains, we retain the lightweight MLP structure for efficiency. **(II)** Next, we evaluate several connectors for the refinement phase (SeR). Among them, combining our base configuration (a) with structure (A), which utilizes both proposed inputs, yields strong performance. Although we also applied (A) to configuration (b), the improvement was marginal, so we did not adopt it as our baseline. **(III)** Finally, we investigate the impact of selecting

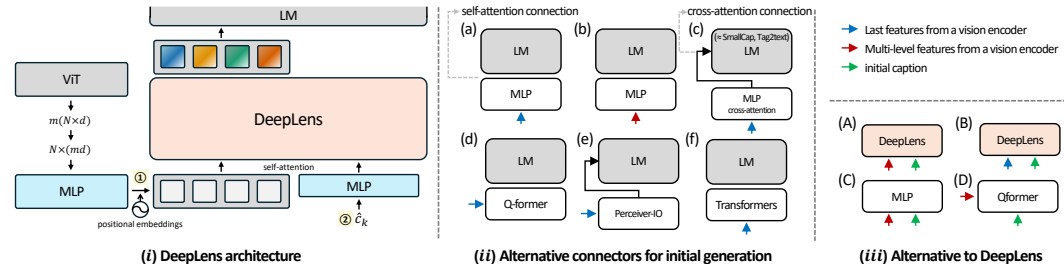

**(i)** DeepLens architecture     **(ii)** Alternative connectors for initial generation     **(iii)** Alternative to DeepLens

Figure 11: **(i)** Illustration of the DeepLens architecture. This is designed for integrating ① token embeddings from the initial caption and ② multi-layer visual features for refinement. **(ii)** and **(iii)** Ablation study of multimodal connector designs. Each configuration corresponds to evaluation in Table 10.

| **(I)** connectors for initial generation | | | | | |
|---|---|---|---|---|---|
| (a) | (b) | (c) | (d) | (e) | (f) |
| 129.6 ✓ | 130.6 | 125.9 | 127.7 | 122.9 | 130.9 |

| **(II)** (a) + connectors for SeR | | | | (b) + con. |
|---|---|---|---|---|
| (A) | (B) | (C) | (D) | (A) |
| 133.5 ✓ | 131.9 | 132.1 | 131.9 | 133.6 |

| **(III)** indexes of selected layers in ViT | | | | |
|---|---|---|---|---|
| {23} | {13, 23} | {15, 23} | {15, 19, 23} | {13, 18, 23} |
| 131.9 | 133.0 | 133.0 | 133.3 | 133.5 ✓ |

Table 10: Ablation results evaluating connector types, refinement configurations, and ViT layer selections in DeepLens. Architectural variants are illustrated in Figure 11 (b) and (c).

different sets of layers from the ViT encoder. We test multiple combinations and find that using a diverse set of layers {13, 18, 23} results in the highest performance.

# D  DISCUSSIONS

## D.1  WHY DO PREVIOUS SMALL MODELS FALL SHORT?

The development of captioning models can be divided into two eras: before and after the adoption of LLMs. Prior to the integration of LLMs, many approaches relied on architectures that did not involve a significant number of parameters. For example, CaMEL [6] and SmallCap [59] employed GPT-2 models with 125M to 350M parameters, while Tag2Text [25] and LoCCa [72] used BERT models ranging from 300M to 900M parameters. The emergence of models such as ClipCap [52], BLIP [38], and LLaVA [44] shifted the direction of the captioning field. With growing interest in the impressive capabilities of LLMs, much of the research has focused on building MLLMs.

In contrast to this trend, we analyze small captioning models and uncover the following: Aforementioned small captioning models typically adopt an architecture where visual features are injected via cross-attention. Unfortunately, when using a small language model, this design does not appear to be particularly beneficial. Empirically, this limitation is reflected in the performance of structure (c) in Figure 11, which yields a relatively low score of 125.9 as shown in Table 10. Had the field not been so heavily influenced by the rise of LLMs, such limitations in small model architectures might have been identified and addressed earlier.

We adopt the LLaVA architecture and inject visual features into the self-attention inputs, as shown in Figure 11 (a). We show that this simple setup surprisingly delivers stronger performance (i.e., Note 1 in Section 3) and contributes to demonstrating that small models can be viable for practical applications. We hope this encourages a shift toward reducing dependence on LLMs in specific tasks like captioning, and sparks further interest in the development of lightweight yet effective models.

## D.2  LIMITATIONS OF EXISTING MULTIMODAL LARGE LANGUAGE MODELS

We examine the broad challenges faced by existing multimodal large language models (MLLMs), particularly in addressing the issue of visual blindness. Prior works such as Eyes Wide Shut [66]

and Cambrian [65] have identified and attempted to mitigate this issue through the use of multiple vision encoders, including DINOv2 [54], SigLIPv2 [68], and CLIP [57]. However, as illustrated in Figure 12, even models with large parameter sizes continue to exhibit difficulty in generating consistent long-form captions in complex, multi-object scenes. We also conduct experiments using two recent capable MLLMs, LLaVA-Next [37] and LLaVA-OneVision [35]. Despite incorporating the sophisticated approach which is to split an image into grids and utilize all the features from each grid, both models continue to suffer from incorrect captioning results.

These findings suggest that visual blindness is a persistent limitation across model scales and architectures. In this context, we believe that our Sharp-Eyed Refinement framework—which encourages attention to key regions via initial captions and leverages multi-level features from a single vision encoder—is a meaningful and efficient step toward addressing this challenge.

### D.3 Limitations of lightweight captioners

While we have demonstrated the potential of small models in captioning tasks, the use of lightweight LMs unavoidably introduces some limitations. In particular, we observe that the model occasionally suffers from issues such as repetitive phrasing, reduced fluency, limited OCR capability, and a lack of general world knowledge. Examples of these cases are provided in Figure 13. These limitations may stem from two primary factors: (*i*) the small number of parameters, which can restrict the model's capacity for complex reasoning and language generation [30], and (*ii*) the limited scale and quality of training data, as our model was trained on approximately 500K image-caption pairs from ShareGPT-4V [11] which contains machine-generated captions. A natural direction for future work is to investigate how far the capabilities of small models can be scaled with access to *larger and higher-quality training datasets*. In addition to the results in Section D.2, we observe similar issues in larger models such as LLaVA-1.5, suggesting that these challenges remain unresolved [5,66] and require deeper investigation.

### D.4 Limitations of existing evaluation methods

To ensure fair comparison across captioning models, we adopt seven evaluation metrics: BLEU@4 [56], METEOR [16], CIDEr [71], BERTScore [94], CAPTURE [18], CLAIR [8], and MLLM-as-judge [9] (i.e., GPT-based evaluation). For CLAIR and MLLM-as-judge, we randomly select 100 image samples and evaluate them using 10 different seeds to compute both mean and standard deviation. The values reported alongside the $\pm$ symbol indicate the standard deviation across seeds. Despite these efforts, we find that current evaluation methods do not always align well with human judgment. For instance, certain models perform better under specific metrics but worse under others, leading to inconsistent rankings. Moreover, small specialists including ours sometimes exhibit deficiencies in fluency, as discussed in Section D.3, which existing metrics struggle to capture. As we examined, MLLM-as-judge yields relatively reliable results. In contrast, CLAIR showed higher variance across runs, limiting its consistency.

These observations highlight the need for further research into robust captioning evaluation. In particular, future work should account for multiple dimensions of quality, including *fluency, coherence, faithfulness, relevance, informativeness, and completeness*. Moreover, moving beyond traditional n-gram-based matching, reliable evaluation methods powered by MLLMs (e.g., OpenAI GPT) are needed. We believe such advancements can support greater trust in captioning-based applications.

### D.5 Impact and Ablation of Pseudo-Initial Captions in Stage-2 Fine-tuning

We first provide a discussion, with examples, on how the inclusion of pseudo-initial captions leads to better supervision during training. We highlight the following points: (*i*) The model is trained to generate the ground-truth (GT) caption given both the image and the pseudo-initial caption as input: $I + \hat{c} \rightarrow$ Our model $\rightarrow c$. (*ii*) If the pseudo-initial caption is generated following the strategies in Section F.1, then it is unlikely to include too many differing tokens from the GT caption.

Suppose we have "A woman in a room with a cat." as the $c$ and $\hat{c}$ in Table 11.

In such cases, if each pair of pseudo-initial caption and GT caption contains a reasonable number of overlapping tokens, the model is encouraged to refer to the pseudo-initial caption and selectively revise the differences to produce the GT caption. Conversely, if the pair contains too many differing

| Model | Eyes Wide Shut CVPR24 | Cambrian NeurIPS24 | LLaVA-Next arXiv24 | LLaVA-OneVision TMLR25 | SmolVLM arXiv25 |
|---|---|---|---|---|---|
| Instruction | "Describe the photo within 55 words." | | | | |
| Toward solving "visual blindness" | Using CLIP + DINOv2 together | Using CLIP + DINOv2 + SigLIP + ConvNeXt-XXL together | Splitting an image into grids and utilizing all the features from each grid | Splitting an image into grids and utilizing all the features from each grid | - |
|  | The image features a large brown bear sitting on a rocky surface, surrounded by green moss. The bear appears to be looking at the camera, with its head tilted to the side. The bear's body is covered in fur, and it seems to be in a relaxed position, possibly resting or observing its surroundings.<EOS> | A brown bear sits on a moss-covered rock, surrounded by green foliage, in a forest setting.<EOS> | A brown bear is standing on a rock. The bear is looking at the camera. The rock is covered in green moss. The bear is in a forested area. The bear is the only animal in the photo. The bear is standing on a rock | A brown bear is sitting on a moss-covered rock, surrounded by greenery and trees.<EOS> | In the center of the image there is a bear sitting on a rock. In the background there are trees and plants.<EOS> |
|  | The image features a small harbor with several boats docked at the pier. The boats are of various sizes and are lined up along the dock. The harbor is surrounded by houses, giving it a quaint and charming atmosphere. The sky above the harbor is cloudy, adding to the serene ambiance of the scene.<EOS> | The image captures a serene scene of a canal lined with traditional Dutch houses. The houses, painted in a variety of colors, feature steep roofs and are adorned with gabled facades, a characteristic feature of Dutch architecture. Several boats are docked along the canal, adding to the charm of the scene. The sky above is a clear blue, and the water in the canal reflects this hue, creating a tranquil atmosphere. The perspective of the image is from the water, looking towards the houses, giving a sense of being on the canal and looking towards the charming houses. The image does not contain any text. | The photo captures a serene scene of a canal, where several boats are docked. The boats are lined with quaint houses, their red roofs contrasting with the blue sky. The image exudes a sense of tranquility and harmony with nature. The water is calm, reflecting the peacefulness of the scene. | The image shows a serene harbor with several sailboats docked at the pier. The boats have tall masts and are moored in calm waters. In the background, there are quaint houses with red roofs and white walls, creating a picturesque coastal scene under a clear blue sky. | This image is taken outdoors. In the middle of the image, there is a body of water, which appears to be a river or a lake. On the left side of the image, there is a building with a white roof and a few windows. On the right side of the image, there is a building with a red roof and a few windows. In the background, there are many houses with different colors and designs. The houses have a lot of windows and roofs. There is a fence along the side of the houses. |
|  | The image features a desk with two computer monitors, a keyboard, and a mouse. The monitors are placed side by side, with one being larger than the other. A bottle is also visible on the desk, located near the center. The desk is situated in an office setting, with a TV mounted on the wall behind the monitors. | The image captures a busy office desk, featuring three computer monitors, each displaying different content. The leftmost monitor shows a news article, the center one displays a black and white image, and the rightmost presents a complex pattern. A keyboard and mouse are positioned in front of the monitors, ready for use. The desk is cluttered with various items, including a phone, a notepad, and a pen, suggesting a workspace in active use. The background is adorned with a bulletin board, filled with papers and notes, adding to the overall impression of a bustling, productive environment. | The photo shows a cluttered desk with two computer monitors, a keyboard, and a mouse. The monitors display various text and images, including what appears to be a news headline and a photograph. The desk is also adorned with papers, a bottle, and a small figurine. The background features a bulletin board with notes and pictures. | The image shows a cluttered office desk with two computer monitors, a keyboard, and a mouse. There are various papers, a bottle of hand sanitizer, and a small red object on the desk. The background features a bulletin board with several notes and pictures pinned to it. | This is a picture of a desk with monitors, keyboard, mouse, papers, and some objects on it.<EOS> |

Figure 12: Despite their size, existing MLLMs struggle with visual blindness, especially in complex scenes. We hope our approach offers a meaningful step toward alleviating this issue.

| | Generalist (LLaVA-1.5-7B) | Our lightweight specialist (450M) | Category |
|---|---|---|---|
|  | The image features a large brown bear sitting on a **moss-covered** rock in a forest. The bear appears to be relaxed and enjoying its surroundings. The **moss-covered** rock provides a natural habitat for the bear, and the forest setting creates a serene and peaceful atmosphere. | A majestic brown bear rests on a **moss**-covered gray rock in a lush forest. The bear's head is turned to the left, surrounded by vibrant green **moss** and **mossy** rocks, creating a serene and harmonious natural scene. The bear's fur blends with the moss, enhancing its majestic appearance. | Repetitive expression |
|  | The image features a harbor with several **boats** docked in the water. The **boats** are of various sizes, and they are lined up along the dock. The harbor is situated near a town, as evidenced by the presence of houses and buildings in the background. | A picturesque harbor scene with a variety of boats, including a large **sailboat** and a smaller **sailboat**, docked in front of a row of houses with red tile roofs. The houses have a mix of white, gray, and brown roofs, and some have balconies and windows. The water is calm and reflects the blue sky above. | Repetitive expression |
|  | The image features a desk with two computer monitors on it. One of the monitors is **turned off**, while the other is turned on. A keyboard is placed in front of the monitors, and a mouse can be seen on the right side of the desk. | A cluttered desk in an office features two computer monitors, a keyboard, and a mouse. The left monitor displays **a black and white** image, while the right displays **a black and white** abstract design. A bulletin board and a bookshelf are visible in the background. | Repetitive expression |
|  | The image depicts a spacious living room with wooden floors and high ceilings. The room is filled with furniture, including a couch, a chair, and a dining table. There is a large window in the room, allowing natural light to fill the space. | A beautifully designed living room showcases a high ceiling with wooden beams and a high ceiling with a skylight. A plush beige sofa and a wooden coffee table with a  rug create an inviting atmosphere, complemented by natural light from a large window. | Lack of fluency |
|  | A woman in a white dress is playing tennis, swinging her racket to hit a tennis ball. | A dynamic scene features a female tennis player in a white outfit and cap, poised to hit a yellow ball with her blue racket. The **blue** and white racket contrasts with the **red** court, while a **green** fence and a **blue** wave create a captivating backdrop. The image captures the intensity of the moment. | Lack of fluency |
|  | The image is a close-up of a white camera, possibly a Polaroid, sitting on top of a leaf-shaped object." | A white camera with a black camera lens is placed on top of a white palm frond. The frond has a curved shape and is surrounded by a light blue background. | Lack of fluency |
|  | The image is a map of Honduras, a country located in Central America. The map shows the country's borders, as well as its major cities, such as Tegucigalpa and San Pedro Sula. The country is surrounded by other Central American countries, including Guatemala, El Salvador, and Nicaragua | A comprehensive map of Mexico is presented, highlighting the country's geographical layout. The map includes the country's name in green and the country's name in gray, with a white background for clarity. The map is bordered by a black border." | Lack of OCR capability |
|  | A pair of black Dr.Martens boots is displayed on a white background. The boots are positioned next to each other, showcasing their design and quality | A pair of stylish **black leather boots** with gold accents is displayed side by side on a white background. The image highlights the contrast between the boots and the white background | Limited world knowledge |

Figure 13: Despite their efficiency, small models suffer from various limitations. Further research is required to assess how high-quality data can compensate for their weaknesses.

tokens, the model is likely to avoid relying on the pseudo-initial caption and instead generate the GT caption independently. This mechanism helps the model refine incorrect parts rather than regenerate the caption from scratch, thereby mitigating misleading supervision.

| pseudo-initial caption $\hat{c}$ | different tokens $E = t \mid \hat{c}(t) \neq c(t)$ in Section 5 |
|---|---|
| A woman in a room with two dogs | two / dogs |
| A cat sitting on a chair in front of the window. | sitting / on a chair / in front of the window |

Table 11: Token-level differences between pseudo-initial and ground-truth captions.

| Detailed captioning | CIDEr [71] | gain | CAPT [18] | gain | GPT [9] | gain |
|---|---|---|---|---|---|---|
| Our specialist fine-tuned via Stage 1 | 40.5 | - | 45.9 | - | 2.74±0.12 | - |
| finetuned w/Data1 | 42.5 | +2.0 | 47.2 | +1.3 | 2.86±0.11 | +0.12 |
| finetuned w/Data2 | 41 | +0.5 | 46.6 | +0.7 | 2.76±0.12 | +0.02 |
| finetuned w/Data3 | 43.6 | +3.1 | 48.4 | +2.5 | 3.02±0.12 | +0.28 |
| finetuned w/Data4 | 43.8 | +3.3 | 48.2 | +2.3 | 3.04±0.09 | +0.30 |

Table 12: Detailed captioning results on ShareGPT4V [11] & DCI [70] with fine-tuning on four types of pseudo-initial captions. The comparison highlights how different levels of overlap or inclusion of GT captions affect model performance.

As an additional experiment, we fine-tune our captioner via Stage 2 as in Figure 7 using pseudo-initial captions generated under the following four conditions: (Data 1) Initial captions generated by the captioner fine-tuned via Stage 1, (Data 2) Pseudo-initial captions that are significantly different from the GT caption, (Data 3) Pseudo-initial captions with minor modifications from the GT caption, which is our original strategy, and (Data 4) Pseudo-initial captions consisting of (i) one generated as in Data3, and (ii) the GT caption itself (e.g., GT: "A woman in a room with a cat"; pseudo-initials: "A boy in a room with a dog" and "A woman in a room with a cat").

Table 12 demonstrates that Data 1 provides reasonable gains within our framework, while Data 2 leads to the anticipated issue where the model tends to ignore the initial caption and generate a new one. Interestingly, Data 4 performs comparably to Data 3, and together with the results from Data 1, indicates that our framework is *robust* to moderate variation in pseudo-initial caption quality.

## D.6 FURTHER RESEARCH QUESTIONS

To guide future exploration, we outline several research directions that may advance the field of lightweight captioning and multimodal learning more broadly:

1. What is the minimal model size required for a captioning specialist to be practically useful in real-world assistive technologies? Where does the performance-efficiency trade-off stabilize?

2. Although supervised fine-tuning is commonly used in multimodal training, recent work has shown the benefits of reinforcement learning methods in the LLM field [67]. Can similar reward-based training be applied to enhance captioning models?

3. The LLaVA framework remains a popular baseline due to its simplicity and open-source accessibility. However, could a unified architecture, such as a fully integrated Transformer-based multimodal model, serve as a better foundation for building practical captioners?

4. Multimodal models heavily rely on the quality of the representations produced by vision encoders. To what extent can we trust visual features from models like SigLIP [90] or MAE [92], especially for fine-grained captioning tasks?

5. Similar to self-refinement in LLM research [51], how can we enable our framework to iteratively improve its own outputs?

6. Can our framework be effective in simple VQA settings? Is an LLM necessary in such setups?

7. Do potential biases (e.g., gender, occupation, and age) exist in captioning models, and if so, how can we address them together with cultural contexts and deployment domains such as indoor, outdoor, and robotic environments? Could domain adaptation with user feedback serve as a solution?

We consider these questions promising directions for future research.

## E ADDITIONAL EXPERIMENTAL RESULTS

**Analysis of captioning operation.** We present additional results from the analysis experiments conducted in Section 3, as shown in Figure 14. These results support our earlier findings that single-

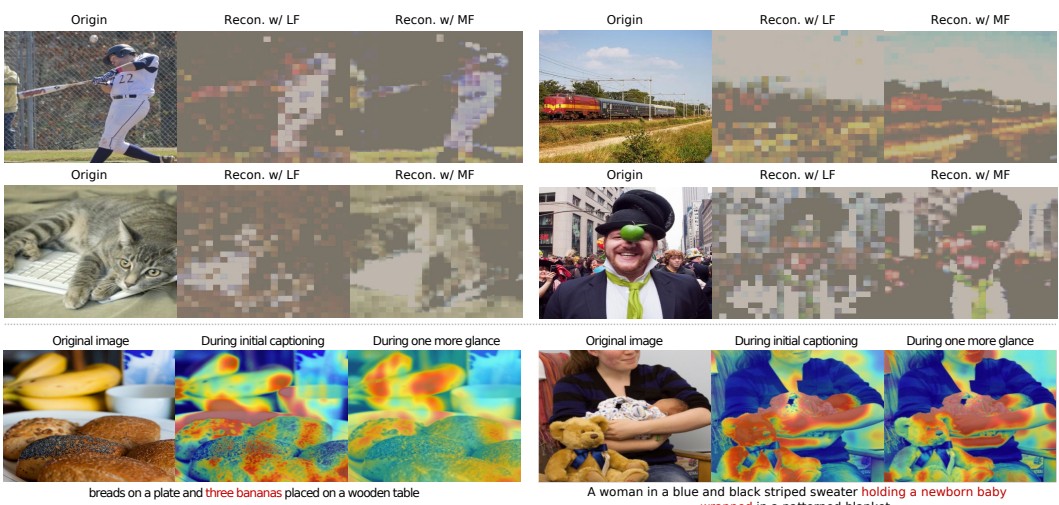

Figure 14: Additional results corresponding to the analyses in Figure 5 and Figure 6. It illustrates diffuse attention patterns (top) and the limited visual detail captured by CLIP features (bottom).

| GT caption | pseudo-initial caption |
|---|---|
| A woman in a room with a cat | A woman in a room with two cats |
| | A woman in a room with a dog |
| | A woman in a room with a cat |
| A multicolored motorcycle rests outside of a sheep farm | A multicolored bicycle rests inside a sheep farm |
| | A bright red motorcycle rests outside of a sheep farm |
| | A multicolored motorcycle races around a sheep farm |

Table 13: Few-shot examples of GT captions and corresponding pseudo-initial captions.

pass captioners tend to adopt diffuse attention patterns, and that CLIP's final-layer features may lack the granularity needed for fine-grained description.

**Comparison between our specialist and LLaVA-1.5-7B.** Additional results comparing our lightweight specialist with the large multimodal generalist (LLaVA-1.5-7B), as discussed in Section 3, are shown in Figure 15. Despite its limited parameter size and a frozen vision encoder, our specialist demonstrates unexpectedly strong performance on captioning benchmarks, indicating that small models may still handle detailed captioning tasks reasonably.

**Impact of Sharp-Eyed Refinement.** Additional results related to our refinement framework are presented in Figure 16, building on the findings discussed in Section 6. Our refinement method yields caption-quality improvements in both single-sentence and detailed captioning.

# F    EXTENSIVE DETAILS ON THE DATASET

## F.1    HOW TO GENERATE THE PSEUDO-INITIAL CAPTION $\hat{c}_k$?

We describe our strategy to generate the pseudo-initial caption as the following. The instruction prompt used for this strategy is shown in Figure 18.

- Pseudo-initial captions are generated by modifying 0–3 points from the ground-truth (GT) caption.
- Modifications are based on four categories: Entity (e.g., chair, cat), Attribute (e.g., color, material like wooden, count like three cups, texture, shape, size, inspired by DSG [1]), Relation (e.g., A in front of B, or B in front of A), Action (e.g., eating, blowing).
- Except the modifications, the overall sentence structure and style should be preserved.
- It can occasionally be identical to the GT caption.
- Few-shot examples are provided like the examples in Table 13.

We consistently generate three pseudo-initial captions per GT caption. Hence, the dataset can be summarized as Table 14, where '#pairs' indicates the number of image-caption pairs.

| Dataset | #images | #GT captions per image | #pairs for fine-tuning stage 1 | #pseudo-initial captions per GT caption | #pairs for fine-tuning stage 2 |
|---|---|---|---|---|---|
| COCO [14] | 113K | 5 | 565K | 3 | 1.6M |
| ShareGPT4V [11] | 100K | 5 | 500K | 3 | 1.5M |
| DCI [70] | 7.4K | 10 | 74K | 3 | 0.2M |
| GLaMM [61] | 550K | 1 | 550K | 3 | 1.6M |

Table 14: Statistics of datasets used for fine-tuning. Notably, the number of images in fine-tuning stages 1 and 2 are *identical*.

## F.2 DATASETS FOR DETAILED CAPTIONING

While MS COCO [14] has long served as the standard benchmark for image captioning, its annotations are typically limited to single-sentence descriptions that often fail to capture the richness of visual content. Recent works [88,21,41,18] have highlighted this limitation and emphasized the need for more detailed captioning datasets. In this study, we leverage three datasets that offer relatively high-quality and more comprehensive image descriptions: ShareGPT-4V [11], DCI [70], and GLaMM [61]. For ShareGPT-4V, captions are initially generated by GPT-4o and subsequently refined by human annotators. DCI contains human-authored captions. Finally, GLaMM generates detailed captions by aggregating outputs from numerous open-source tools, including object detectors and scene parsers, and composing them using an LLM.

## F.3 SHAREGPT & DCI

The DCI dataset [70] contains 7.4K training images and 0.4K test images, with each image paired with 10 human-written captions averaging 55 words in length.

The ShareGPT4V dataset [11] includes 100K images, each accompanied by a single long and human-verified caption of approximately 200 words. Using this format directly poses a challenge for fair evaluation with n-gram-based metrics, which benefit from having multiple reference captions per image. So, we summarized the original caption into five captions, each containing roughly 50 words, since generating multiple captions while preserving the original length could introduce hallucinations. The prompt used for this summarization process is presented in the next section, and the resulting dataset will be made publicly available.

Given the relatively small scale of DCI compared to MS COCO (which contains 118K images with five captions per image), we combined DCI with the processed ShareGPT4V dataset to construct a unified benchmark. This yielded 102.4K training images (100K from ShareGPT4V and 7.4K from DCI), each with ten or five detailed captions, and 5K test images—comparable in scale to MS COCO.

## F.4 GLAMM

The GLaMM dataset [61] contains automatically generated captions produced using a combination of object detection, scene-graph parsing models, and LLMs. On average, each caption consists of approximately 45 words. In our experience, despite GLaMM's reliance on extensive visual tools, the quality of the generated captions was often inconsistent. Frequent factual inaccuracies were observed, for example, one to two incorrect words appearing in every two and three captions. Each image in GLaMM is paired with a single caption, which presents challenges for n-gram-based evaluation, as previously discussed. Nevertheless, we include this dataset in our experiments. We randomly sampled 600K image-caption pairs from the full dataset, allocating 30K for testing and using the remaining 570K for training. Possibly due to the quality issues mentioned above, models trained on GLaMM generally underperformed compared to those trained on ShareGPT4V and DCI.

## G PROMPT TEMPLATES

We share the prompt templates used when interacting with OpenAI's GPT models throughout our study. First, the prompt used for summarizing long captions into shorter ones, introduced in

| | Pretraining | Fine-tuning |
|---|---|---|
| Dataset | LCS-558K [43] | MS COCO [14] or SharedGPT [11] + DCI [70] |
| Adapter | 4-layer MLP with GELU | 4-layer MLP with GELU / Deeplens |
| Trainable | Adapter layers only | Adapters + Language Model |
| Training Epochs | 1 | 10 / 2 |
| Learning Rate | $1 \times 10^{-4}$ | $2 \times 10^{-5}$ |
| Weight Decay | 0 | 0 |
| Warm-up Ratio | 0.03 | 0.03 |
| Learning Rate Scheduler | Cosine decay | Cosine decay |

Table 15: Hyperparameters used for model training. The settings to the left and right of the **/** correspond to those used in Section 4 and Section 3, respectively.

| LLaVA-1.5 | CIDEr [71] | BERTScore [94] | CAPT [18] |
|---|---|---|---|
| "Describe the photo within 55 words" | 36.1 | 36.6 | 48.6 |
| "Describe the photo in detail" | 12.8 | 17.6 | 40.6 |

Table 16: Performance of LLaVA-1.5 under different prompt instructions.

Section F.3, is shown in Figure 17. Second, the prompt used to generate pseudo initial captions for our refinement framework, as described in Section 5, is provided in Figure 18. The examples generated using it are in Figure 20. Lastly, the prompt used in the MLLM-as-judge evaluation setup is shown in Figure 19. We note that we strictly follow the template proposed in the original MLLM-as-judge papers [8,9].

# H  EXPERIMENTAL DETAILS

## H.1  PRETRAINING AND FINETUNING

Our implementation is initialized from the LLaVA-1.5 repository [43]. To reduce reliance on LLMs, we replace LLaMA with the OPT series of language models [93]. The training process consists of three stages: pretraining, finetuning for caption generation (in Section 3), and finetuning for sharp-eyed refinement (in Section 5). The hyperparameters used for each stage are summarized in Table 15. We closely follow the original LLaVA training setup. As emphasized in Section D.1, our method differs from previous small-model approaches in that the visual features are directly used as self-attention inputs to the language model.

## H.2  EXPERIMENT SETTING OF LARGE MULTIMODAL GENERALIST

Table 2 compares specialist models with several generalist MLLMs, including InstructBLIP [15], Unified-IO-XL [47], Shikra [10], Qwen-VL [3], and LLaVA-1.5 [43]. These generalist models were reported to be trained on MS COCO images, but not on datasets such as ShareGPT4V, DCI, or GLaMM. Instead, they were instruction-tuned on large-scale datasets; for example, Qwen-VL and InstructBLIP were trained with approximately 1.5B and 130M instruction samples, respectively. For this reason, we categorize them as generalist models.

The generalist MLLMs were evaluated using publicly available checkpoints from their official GitHub repositories, without any additional fine-tuning. For the single-sentence captioning task, we used the prompt: *"Provide a one-sentence caption for the provided image."* For the detailed captioning task, we used the prompt: *"Describe the photo within 55 words.".* Additionally, we conduct an experiment on LLaVA-1.5 with the alternative instruction, as shown in Table 16. These results reveal two important points: (1) evaluation metrics such as BERTScore [94] penalize long-form outputs, and (2) longer generations tend to induce more hallucinations, consistent with prior observations [26].

## H.3  ATTENTION MAP VISUALIZATION

As part of our analysis in Section 4, we evaluate whether the model attends to the appropriate image regions when generating specific words in a caption. To this end, we adapt the visualization code provided by API [89], originally designed to point out attention maps between images and questions in VQA tasks. We modify the attention hooking module (https://github.com/yu-rp/apiprompting/blob/master/API/API_LLaVA/hook.py) to visualize attention between

| Optimizer | Learning Rate | $\beta$ | Weight Decay | Batch Size | Epoch |
|-----------|---------------|---------|--------------|------------|-------|
| AdamW | 1e-5 | (0.9, 0.95) | 0.05 | 64 | 10 |

Table 17: Hyperparameters used for training the MAE decoder in the image reconstruction experiment on MS COCO.

image regions and selected words within captions. This allows us to inspect which parts of the image the LM focuses on when producing certain tokens, such as the phrase 'a white toilet' in Figure 5.

### H.4   IMAGE RECONSTRUCTION

To investigate whether CLIP's visual representations are coarse or ambiguous, we conducted an image reconstruction experiment using a Masked AutoEncoder (MAE) framework in Section 4. In this setup, a ViT encoder produces visual features, which the decoder then uses to reconstruct the image. We adopt the same visual encoder as used in LLaVA-1.5, 'CLIP ViT-L/14-336', and keep its parameters frozen. The decoder receives the visual embeddings without masking and predicts the corresponding RGB image. We utilize two types of visual inputs: (i) last-layer features (*i.e.*, LF), and (ii) multi-level features (*i.e.*, ML), where ML consists of outputs from layers 13, 18, and 23 of the encoder. The decoder is trained on 100K images from the MS COCO dataset. Further architectural details are available in the MAE repository, https://github.com/facebookresearch/mae/blob/main/models_mae.py. The hyperparameters for optimizing the decoder are summarized in the table below.

### H.5   LONG RANGE VIDEO QUESTION ANSWERING

To assess the feasibility of deploying lightweight captioners in real-world applications, we evaluate our model on the Long-Range Video Question Answering task [91]. This task requires the system to answer multiple-choice questions based on user queries about videos that are 10 minutes or longer in duration. We emphasize that there are currently *no video MLLMs* capable of handling this task directly. Most existing models [34,4,97] impose limits on the number of input video frames they can process, making it difficult to cover the full temporal span of long videos.

To address this, recent approaches [86, 75] propose first extracting per-frame captions and then injecting both the captions and the question into an LLM, which can handle over 100K tokens in a single prompt. For evaluation, we follow the setup provided in the official implementation of LLoVi (https://github.com/CeeZh/LLoVi), replacing only the captioning models. When using the generalist model, we generate a caption for each frame using the prompt: "Describe this frame within 50 words". For our specialist model, we use the same prompt and apply the version fine-tuned on the ShareGPT4V [11] and DCI [70] datasets.

## I   LLM USAGE

We used a large language model (LLM) solely to aid in polishing the writing, such as improving grammar, clarity, and fluency of the manuscript. The LLM was not involved in research ideation, implementation, or analysis, and all scientific contributions are entirely our own.

| | Generalist (LLaVA-1.5-7B) | Our lightweight specialist (450M) | | Generalist (LLaVA-1.5-7B) | Our lightweight specialist (450M) |
|---|---|---|---|---|---|
| | Two zebras are standing in the grass. | A black and white photo of a zebra standing in the grass. | | A blue and white airplane with the number 2 on it. | A blue and white airplane on a grass field. |
| | A man in a brown jacket is using a computer. | A group of people sitting at a desk with computers. | | Two canoes are sitting on the beach. | Two kayaks on the beach next to water. |
| | A pole with street signs on it. | A pole with a green and white sign on it. | | A variety of doughnuts are displayed in a bakery. | A display case filled with donuts and pastries. |
| | A man wearing a banana costume is smiling. | A man dressed in a costume with bananas on his head. | | A basketball player with the number 12 on his jersey. | A young man holding a basketball in his hands. |
| | Three people walking in the rain holding umbrellas. | A group of people walking in the rain with umbrellas. | | A bird flies over a body of water on a cloudy day. | A black and white photo of birds flying over a body of water. |
| | Three zebras and wildebeest grazing in a field. | Three zebras grazing in a grassy field. | | A black and brown dog eating food off of a plate. | A dog standing on the floor eating a plate of food. |
| | A man is riding a motorcycle on the street. | Two men riding a motorcycle down a street. | | A large group of people are gathered in front of a clock tower. | A crowd of people are gathered around a clock tower. |

| | Generalist (LLaVA-1.5-7B) | Our lightweight specialist (450M) |
|---|---|---|
| | The image features a black cow standing in a grassy field near a fence. The cow appears to be grazing on the grass, and it is tied to the fence with a rope. The fence encloses the area where the cow is grazing, providing a sense of security for the animal. | A black cow grazes peacefully in a lush green field, its head lowered as it faces the camera. A rustic red brick wall and a white house with a red roof are visible in the background, creating a serene rural atmosphere. The cow's calm demeanor adds to the tranquil scene. |
| | In the image, there is a man wearing a black jacket and blue jeans, standing on a sidewalk. He is holding a motorcycle helmet in one hand and a baby stroller in the other. There are several other people in the scene, including a woman taking a picture with her cell phone. | A casually dressed man with curly hair walks along a gravel path in a park, holding a black helmet and a stroller. The scene is filled with people, including a woman taking a photo and a man with a stroller, all set against a backdrop of lush greenery. A stone building looms in the distance. |
| | Two men are standing on a soccer field. One of the men is wearing a red jacket, while the other is wearing a blue jacket. Both men are wearing blue pants. | A grassy field serves as a backdrop for two men, one in a red jacket and the other in a blue jacket. Both men are focused on their tasks, with the man in the red jacket on the left and the man in the blue jacket on the right. |
| | The image is a close-up of a plaid pattern, showcasing the intricate design of the fabric. The plaid pattern consists of alternating blue and green squares, creating a visually appealing and classic design. | A visually striking image of a plaid pattern features a grid of green and black squares, arranged in a grid of four rows and four columns. The plaid's uniformity and uniformity create a harmonious and visually appealing composition. |
| | The image features a large, old cannon sitting on a brick platform. The cannon is positioned in the middle of the platform, and it appears to be the main focus of the scene. The cannon is quite large, occupying a significant portion of the platform. | A majestic medieval cannon rests on a stone wall, featuring a vibrant green and red color scheme. The cannon is positioned on a wooden platform, facing the ocean, with a clear blue sky in the background, enhancing the scene's tranquility. The image captures the cannon's historical significance. |
| | A black and white penguin is standing on a rock, holding a fish in its beak. The penguin appears to be in the process of eating the fish. | A serene image features a puffin perched on a rock, its black and white body contrasting with its orange feet. The puffin holds a catch of a small fish in its beak, set against a blurred yellow and brown background that enhances the tranquil atmosphere. The bird's red beak and feet add color to the scene. |
| | The image features a close-up view of a piece of cake on a white surface. The cake appears to be made of wafers, giving it a crumbly texture. The piece of cake occupies a significant portion of the image, covering almost the entire frame. | A close-up view of two beige crackers resting diagonally on a white surface. The crackers are arranged diagonally, with the top one slightly higher than the bottom one, creating a sense of depth against the stark white background. The image emphasizes the texture of the crackers. |
| | The image features a black bear sitting on a wooden bench in a park-like setting. The bear appears to be enjoying a snack, as it is eating something off the bench. The bench is positioned in the middle of the scene, with the bear occupying a significant portion of the image. | A black bear sits on a wooden platform in a verdant forest, its head resting on its paws. The platform, made of light-colored logs, is surrounded by lush greenery and bamboo, creating a serene atmosphere that highlights the bear's natural habitat. |
| | A man is standing on skis in the snow, wearing a black jacket and red gloves. He is smiling and appears to be enjoying his time on the slopes. There are several other people in the background, some of whom are also wearing skis. | A skier in a black jacket and red beanie stands ready for action on a snowy mountain slope, holding ski poles. A rustic wooden cabin with a gray roof is visible behind him, while other skiers and snowboarders populate the background under a clear blue sky. Snow-covered trees and a clear blue sky complete the winter scene. |

Figure 15: Qualitative comparison between our lightweight specialist and the large multimodal generalist LLaVA-1.5-7B using MS-COCO [14] (top) and ShareGPT4V [11] & DCI [70] (bottom). Despite its smaller size and simpler architecture, our model produces competitive descriptions.

| | Initially generated captions | After sharp-eyed Refinement | | Initially generated captions | After sharp-eyed Refinement |
|---|---|---|---|---|---|
| | A plate of food on a table at a restaurant. | A white plate topped with a fried fish and potatoes. | | A magazine cover of a soccer team. | A magazine cover of a professional soccer player. |
| | A white toilet sitting next to a bathroom sink. | A bathroom with a wooden shelf and a white sink. | | A couple of giraffe standing next to each other. | A mother giraffe standing next to a baby giraffe. |
| | Two hot dogs with mustard and ketchup on a paper plate. | Two hot dogs with mustard on a paper plate. | | A semi truck is driving down the street. | A white truck driving down a street next to houses. |
| | A couple of men standing next to each other. | A couple of men sitting next to each other. | | A herd of sheep grazing on a lush green field. | A herd of sheep standing on top of a dry grass field. |
| | A man standing on top of a sandy beach near the ocean. | A man standing on the beach watching a flock of seagulls. | | A close up of a zebra eating grass. | a close up of a zebra eating grass near a wall. |
| | An orange cat laying on top of a bed. | A cat walking across a bed in a bedroom. | | A person sitting on a surfboard in the water. | A woman sitting on a surfboard in the ocean. |
| | A wooden bench with a handbag on it. | A wooden bench with a glove on it. | | A person riding a horse along the beach. | A person riding a brown horse along a sandy beach. |
| | A traffic light hanging over a city street. | A green traffic light hanging over a street. | | A red fire hydrant sitting on the side of a road. | A red and blue fire hydrant sitting on the side of a road. |
| | A man and a boy on a surfboard in the ocean. | A man and a young boy riding a surfboard in the ocean. | | A baseball player holding a bat next to home plate. | A baseball player holding a bat on a field. |

| | Initially generated captions | After sharp-eyed Refinement |
|---|---|---|
| | A vibrant red and yellow train travels along a curved track in a verdant landscape, surrounded by lush greenery. Power lines crisscross above, and a calm body of water reflects the clear blue sky, creating a harmonious blend of nature and technology. The image captures a moment of tranquility. | A vibrant red and yellow train travels along a straight track in a verdant landscape, surrounded by lush greenery. Power lines crisscross above, and a calm body of water reflects the clear blue sky, creating a harmonious blend of nature and technology. The image captures a moment of tranquility. |
| | A whimsical scene features a pink pig resting on a wicker basket, its head resting on a magazine. The basket is placed on a black surface, with a white wire mesh wall in the background, creating a warm and inviting atmosphere. | A whimsical scene features a pink ceramic pig resting on a wicker basket, its head turned to the side. The basket is placed on a black surface, with a white wire mesh wall in the background, creating a cozy atmosphere. |
| | An exhilarating moment captures a surfer in a black wetsuit skillfully riding a wave on a white surfboard. Two other surfers are visible in the background, one crouched and the other paddling, all set against a backdrop of a clear blue sky and distant mountains. The image captures the thrill of surfing. | An exhilarating moment captures a surfer in a black wetsuit skillfully maneuvering on a wave, causing a spray of water. Two other surfers are visible in the background, partially submerged in the water, all set against a backdrop of a clear blue sky and distant mountains. The image captures the thrill of surfing. |
| 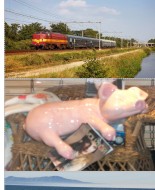 | A vibrant red and yellow steam locomotive is seen pulling a yellow passenger car along a railway track. A worker in an orange vest is visible in the background, surrounded by lush greenery and a clear blue sky. The image captures the essence of a train journey. | A maroon and yellow steam locomotive is seen pulling a series of freight cars along a narrow railway track. A worker in an orange vest is visible in the background, surrounded by lush greenery and dappled sunlight. The image captures the essence of a heritage railway scene. |
| | A whimsical penguin statue, dressed in a vibrant purple and white striped outfit, stands on a beige pedestal in a grassy area. The statue is surrounded by a plaque and a nearby tree, with a serene ocean view visible in the background. The overcast sky adds a soft light to the scene. | A whimsical penguin statue, dressed in a purple and white sports-themed outfit, stands on a paved area near a beige pedestal with a plaque. The statue is positioned in a grassy area with a signpost and bench nearby. The overcast sky adds a soft light to the scene. |
| | A lively urban scene features a red stop sign and a fire hydrant on a wet street. Two people walk along the sidewalk, one holding a red bucket, while graffiti-covered buildings and a train bridge create a vibrant backdrop. | A lively urban scene features a red stop sign and a fire hydrant on a wet street. Three children play near an open fire hydrant, enjoying the strong water flow, while graffiti-covered walls and a train bridge create a vibrant backdrop. |
| | A man in a white t-shirt and blue jeans kneels on the floor, using a white toilet brush to clean the toilet. The beige wall behind him features a red stain, and a green plant adds a touch of nature to the scene. The image captures a moment of cleaning. | A man in a white t-shirt and blue jeans kneels on the floor, working on the toilet with a wrench. The beige wall behind him features a faint stain, and the scene focuses on a plumbing repair rather than cleaning. |
| | A serene moment is captured inside a car, featuring a light tan dog with a blue collar resting on the passenger seat. The dog's head rests on a black backpack, and the window reveals a glimpse of the outside world, enhancing the sense of comfort. The image conveys tranquility. | A serene moment is captured inside a car, featuring a light tan dog with a striped collar resting in the back seat. The dog's head rests on a blue and black bag, and the surrounding shadows enhance the sense of comfort. The image conveys tranquility. |
| | A vibrant yellow butterfly rests on a textured brown surface, its wings spread wide and wings slightly spread. The butterfly's head is turned slightly to the left, showcasing its striking green body and brown wings. The image is captured from a slightly elevated angle. | A vibrant yellow-green butterfly rests on a textured brown surface, its wings spread wide. The moth's head is turned slightly to the right, showcasing its striking green body and brown-edged wings. The image is captured from a slightly elevated angle. |

Figure 16: Qualitative examples of our sharp-eyed refinement. The results show improved caption quality after refinement, using MS-COCO [14] (top) and ShareGPT4V [11] and DCI [70] (bottom).

---

**Prompt to summarize descriptions for ShareGPT4V**

You are an expert in image description. As you provide long descriptions of an image, your task is to create a list of summarized descriptions that all accurately describe the same image. The elements you should keep in mind are as follows:
1) From the given long description, each description must be concise yet comprehensive without creating any hallucinations and must adhere to a 35-50 word limit.
2) Each description should offer a slightly different perspective on the entire image, as shown in the examples below.
3) Like the examples below, each sentence should start with A, An, Two, or similar words, providing a description of the entire image.
4) Each description must be in English only, not in any other language.
5) Each sentence should end with a period (.).
Here's an example of summarized captions for an image:

...

<Example Image A>
A-1. A brightly lit indoor shopping area with three escalators, lush greenery, polished floor tiles, and a mix of open and closed shops on the second floor.
A-2. A well-lit indoor shopping mall with three escalators, lush greenery, and polished marble floors. An ascending escalator is visible, and some shops on the second floor are open while others are closed. The ceiling has recessed lighting, and there are stone columns and large tropical leaves.
A-3. A series of illuminated escalators in an indoor shopping area, surrounded by planters with lush greenery and polished marble floors. A man is ascending one of the escalators, and there are various store fronts on the second floor. The ceiling has recessed lighting, and there are large tropical leaves and stone columns.
A-4. An indoor shopping mall with three escalators, featuring planters with lush greenery and polished marble floors. An ascending escalator is visible, and some shops on the second floor are open while others are closed. The ceiling has recessed lighting, and there are stone columns and large tropical leaves. The image highlights the intersection of technology and nature.
A-5. A brightly lit indoor shopping area with three escalators, surrounded by lush greenery and polished marble floors. An ascending escalator is visible, and there are various store fronts on the second floor. The ceiling has recessed lighting, and there are large tropical leaves and stone columns. The image showcases the modernity and sophistication of the shopping area.

<Example Image B>
B-1. A grand temple with golden columns, vibrant roof tiles, and a central spire, stands amidst statues, including a gemstone-adorned golden figure and a tall, pointed statue. The temple is surrounded by marble walls, a tall lamppost, and a small tree. The sky is mostly clear, with a few scattered clouds.
B-2. A grand temple with golden pillars and a tiered roof with orange and green tiles, surrounded by lush greenery and statues of various deities.
B-3. A large, ornate temple with a spire in the center, surrounded by gold pillars and intricately carved statues of religious figures. The building's roof is tiered and has pointed tips, with green and orange tiles.
B-4. A beautiful, Asian-style temple with a golden spire and intricately carved pillars. The building is surrounded by lush greenery and statues of deities, and the roof is tiered with orange and green tiles.
B-5. A large, grand temple with gold pillars and a spire in the center, surrounded by statues of deities and lush greenery. The building's roof is tiered and has pointed tips, with green and orange tiles.

<Example Image C>
C-1. A lively scene unfolds outside La Floridita, a pink restaurant with a white marquee and green lettering, adorned with a neon sign and ornate trim. People in casual attire gather outside, near a parked yellow taxi. Trees line the street, alongside a small, boarded-up hotel. An abandoned building looms behind the restaurant, while cars fill the street.
C-2. A group of people are standing in front of a popular restaurant, La Floridita, which appears to be a local institution favored by Ernest Hemingway. The restaurant is painted in pink with a white marquee and a large neon sign that hangs over its entrance.
C-3. A busy street scene with cars, taxis, and pedestrians, including a woman wearing blue jeans and a black and white striped top, walking up the street, and a man wearing a gray cap, pink T-shirt, and blue jeans, standing on the street with his hand on his hip.
C-4. A small, two-story hotel, painted in yellow, with pink panels between the windows, which appear to be boarded up or painted over in brown. The hotel has a sign over the entrance and a small overhang below.
C-5. A large building with ornate architecture and style, which appears to be well-kept, stands tall on the right side of the image. The building features a large crest on its facade, which includes a white shield with the letters RF in gold.

The examples above show summarized descriptions for different images. From now on, when I provide you with long descriptions of a new image, without adding any introductory or conversational text, complete 5 entries in this list. Present summarized descriptions in the following format:
1. <description>
2. <description>
3. <description>
4. <description>
5. <description>

Figure 17: Prompt used for summarizing long-form captions into shorter, multi-reference captions.

---

**Prompt to generating pseudo initial caption for single sentence captioning dataset**

You are a caption rewriting assistant.
- Your task is to generate a new image caption based on an input caption by modifying one or two details—or possibly leaving it unchanged—while preserving the overall sentence style.
- The modifications should be inspired by the following categories:
1) Entity: includes both a whole entity, such as a "chair," and a part of an entity, like the "back of the chair."
2) Attribute: cover various aspects such as color (e.g., "red book"), type (e.g., "aviator goggles"), material (e.g., "wooden chair"), count (e.g., "5 geese"), texture (e.g., "rough surface"), text rendering (e.g., letters "Macaroni"), shape (e.g., "triangle block"), and size (e.g., "large fence").
3) Relation: describes spatial relationships (e.g., "A next to B"), action relationships (e.g., "A kicks B"), and global properties (e.g., "bright lighting").
4) Action: describes verbs or behaviors, such as "eating" or "blowing."
- Imagine there are two images, A and B. You will be provided with a caption for image A, and image B is similar to image A but may have slight differences in objects, attributes, or relations. Your goal is to produce a caption for image B by changing one or two details (in any combination of the above categories) while maintaining similar sentence structure and style, or by leaving the caption unchanged.
- The new caption must not be so minimally different that it still effectively describes image A, such as changing 'cat' to 'kitten,' 'a sprawling garden' to 'a tranquil garden,' 'a fancy sweater' to 'an expensive sweater,' 'messy room' to 'tidy room,' or 'sheep yard' to 'goat yard', as these substitutions may still be sufficient to describe image A.

...

Example1:
Input:"A view of a messy room, with shelves on the wall."
Output:
1."A view of a messy room, with stairs on the left."
2."A view of a messy room, with paintings on the wall."
3."A view of a bright room, with shelves on the ceiling."

Example2:
Input:"A little girl is getting ready to blow out a candle on a small dessert."
Output:
1."A little girl is getting ready to eat a small dessert."
2."A little boy is getting ready to blow out a candle on a small dessert."
3."A little girl is holding out a sparkler on a small dessert."

Example3:
Input:A woman in a room with a cat."
Output:
1."A woman in a room with a cat."
2."A woman in a room with a dog."
3."A woman in a room with two cats."

Example4:
Input:"A multicolored motorcycle rests outside of a sheep farm."
Output:
1."A multicolored bicycle rests inside a sheep farm."
2."A bright red motorcycle rests outside of a sheep farm."
3."A multicolored motorcycle races around a sheep farm."

...

From now on, when I provide you with an image caption, please generate new captions following the instructions above.
Do not include any additional introductory or conversational text. Present new captions in the following format:
1."<caption>"
2."<caption>"
3."<caption>"

---

**Prompt to generating pseudo initial caption for detailed captioning dataset**

You are a caption rewriting assistant. Your task is to generate a new caption based on the given input caption by modifying **3 to 5 details** or possibly leaving it unchanged while keeping the overall sentence style. The modifications should be inspired by the following categories:
1. **Entity**: This can be a whole entity "chair," or a part of an entity "back of the chair."
2. **Attribute**: This includes aspects such as color (e.g., "blue book"), type (e.g., "aviator goggles"), material (e.g., "wooden chair"), count (e.g., "3 geese"), texture (e.g., "rough surface"), text rendering (e.g., "letters on a sign"), shape (e.g., "round table"), and size (e.g., "large fence").
3. **Relation**: This involves spatial relationships (e.g., "A next to B"), action relationships (e.g., "A kicks B").
4. **Action**: Describes verbs or behaviors, such as "eating," "jumping," or "singing."

Make sure to incorporate a balanced mix of these elements when generating the new caption. Do not focus solely on modifying the entity.

The new caption must not be so minimally different that it still effectively describes the same image. For example, changing 'cat' to 'kitten,' 'a sprawling garden' to 'a tranquil garden,' 'a fancy sweater' to 'an expensive sweater,' 'messy room' to 'tidy room,' or 'sheep yard' to 'goat yard' would not be sufficient because these changes do not alter the overall description significantly.

...

Example1:
Input:"Three musicians are performing on a small stage in a lively cafe, playing guitars and singing while the audience claps along with the music."
Output:
1."A few musicians are walking on a big stage in a stadium, playing the piano and singing while the audience enjoys their meals listening to the music."
2."Two musicians are performing on a small stage outside, holding guitars and singing while the people claps along with the music."
3."Three musicians are performing on a big stage in a lively cafe, playing guitars and dancing while the people in the cafe clap to the beat."

Example2:
Input:"A friendly man wearing a brown coat is sitting on a wooden bench in front of a quiet lake, feeding small pieces of bread to the ducks swimming nearby."
Output:
1."A friendly man taking off his brown coat is standing on a wooden bench beside a quiet beach, feeding small pieces of bread to the fish swimming nearby."
2."Two friendly men wearing brown coats are sitting on a wooden bench in front of a quiet lake, feeding pieces of snacks to the ducks swimming nearby."
3."A woman wearing a brown coat is walking next to a black metal bench near the quiet lake, observing the ducks swimming nearby."

Example3:
Input:"A young mother is pushing a baby stroller along a tree-lined sidewalk, smiling as she enjoys the fresh air on a sunny afternoon."
Output:
1."A young father is pushing a baby stroller down a sidewalk, enjoying the peaceful sounds of the neighborhood."
2."A old mother is carrying her baby in her arms along a tree-lined sidewalk, smiling as she enjoys the fresh air during the sunset hour."
3."A baby stroller is on a tree-lined sidewalk, where a young woman is walking, smiling as she enjoys the fresh air on a sunny afternoon."

...

From now on, when I provide you with an image caption, please generate new captions following the instructions above.
Do not include any additional introductory or conversational text. Present new captions in the following format:
1."<caption>"
2."<caption>"
3."<caption>"

Figure 18: Prompt used to generate pseudo initial captions for Sharp-Eyed Refinement.

---

**MLLM-as-judge (i.e., GPT in our main paper) caption evaluation**

(System Prompt)
You are a helpful assistant proficient in analyzing vision reasoning problems.

(Instruction)
Please examine the provided image attentively and serve as an unbiased judge in assessing the quality of the response from an AI assistant regarding the instruction. You will receive a single response from the assistant to user's instruction.

(Noticement)
Your assessment should identify whether the assistant effectively adheres to the user's instructions and addresses the user's inquiry.
In your evaluation, weigh factors such as relevance, accuracy, comprehensiveness, creativity, and the granularity of the responses.
Do not allow the length of the responses to influence your evaluation.
Do not favor certain names or positions of the assistants. Be as objective as possible.

(Criteria)
Use scores to show the quality of the response. Here is the detailed scoring rubric for evaluating the quality of responses from AI assistants:

Poor (1): The response significantly deviates from the user's instruction and fails to address the query effectively. It shows a lack of relevance, accuracy, and comprehensiveness. Creativity and granularity are absent or poorly executed.
Fair (2): The response addresses the user's instruction partially, with evident shortcomings in relevance, accuracy, or comprehensiveness. It lacks depth in creativity and granularity.
Average (3): The response adequately addresses the user's instruction, showing a fair level of relevance, accuracy, and comprehensiveness. It reflects a basic level of creativity and granularity but may lack sophistication.
Good (4): The response is well-aligned with the user's instruction, demonstrating a high degree of relevance, accuracy, and comprehensiveness. It shows creativity and a nuanced understanding with detailed granularity.
Excellent (5): The response perfectly adheres to the user's instruction, excelling in relevance, accuracy, comprehensiveness, creativity, and granularity.

(Conversation that requires evaluation)
[The Start of User Instruction]
{caption_image}

Describe this image in a single sentence.
[The End of User Instruction]

[The Start of Assistant's Answer]
{candidate_caption}
[The End of Assistant's Answer]

Use '[1]', '[2]', '[3]', '[4]', '[5]' to indicate your evaluate score in the key Judgement.

Your output format is as follows.
"
Score: [?]
Reason:
"

---

**CLAIR caption evaluation**

You are trying to tell if a candidate caption is describing the same image as a reference set of captions.
Candidate caption: {candidate_caption}
Reference set: {references_caption}
On a precise scale from 0 to 100, how likely is it that the candidate is describing the same image as the reference set?

Your output format is as follows:
"
Score: [?]
Reason:
"

Figure 19: Prompt used for MLLM-as-judge evaluation, following the original template from [8, 9].

| Ground Truth captions | Generated pseudo initial captions |
|---|---|
| The man and woman are smiling with a box of donuts. | (1) The man and woman are frowning with a box of pastries.
(2) The man and woman are smiling with a box of cookies.
(3) The woman is smiling with a box of donuts, while the man is laughing. |
| A boy surfing a wave on his surfboard. | (1) A girl surfing a wave on her surfboard.
(2) A boy riding a wave on his skateboard.
(3) A boy surfing a small wave on his surfboard. |
| A black cat and a calico cat in a bedroom. | (1) A white cat and a calico cat in a bedroom.
(2) A black cat and a tabby cat in a cozy bedroom.
(3) A black cat and a calico cat on a bed in a bedroom. |
| Five officers are cutting a large colorful military cake. | (1) Five officers are cutting a small colorful military cake.
(2) Three officers are cutting a large colorful military cake.
(3) Five officers are slicing a large decorated military cake. |

| Ground Truth captions | Generated pseudo initial captions |
|---|---|
| An elevated perspective reveals three people walking down a rain-drenched street with umbrellas, one showcasing a vivid floral design while another is gray. They approach a taxi marked by contrasting black and yellow colors, embodying determination as they traverse the wet asphalt. | (1) A high-angle view shows four individuals strolling along a snow-covered street holding umbrellas, one decorated with a bright polka dot pattern while another is bright red. They near a car highlighted by bold white and blue colors, embodying tenacity as they navigate the icy pavement.
(2) From a bird's-eye view, five people are moving across a misty avenue with umbrellas, one featuring a colorful geometric print while another is completely black. They approach a bus marked with distinctive green and white colors, radiating resolve as they cross the slick surface.
(3) An overhead shot displays two figures walking along a wet cobblestone path, each holding umbrellas—one with a pastel striped pattern and the other in dark blue. They make their way toward a car painted in striking orange and gray hues, exemplifying perseverance as they tread on the glistening ground. |
| In a grassy field, three dogs exhibit their lively spirits: a black and gray Schnauzer and a white and brown Papillon play together, while a tan Chihuahua appears more observant. Their contrasting energies highlight the overall joyful essence of their playful interaction. | (1) In a vibrant meadow, two dogs display their energetic personalities: a golden Bichon Frise and a tri-color beagle frolic together, while a small brown Dachshund stands watchful. Their differing antics emphasize the cheerful atmosphere of their playful engagement.
(2) On a sunny hill, four dogs are showcasing their playful spirits: a brown and white Cocker Spaniel and a cream-colored French Bulldog chase each other, while a cautious black Poodle watches nearby. Their varied temperaments capture the delightful essence of their joyous romp.
(3) In a lush park, five dogs are reveling in their playful nature: a brindle Boxer and a fluffy white Maltese dash around, while a curious tan Terrrier remains close to a tree. Their diverse behaviors enhance the vibrant spirit of their delightful interactions. |
| Two towering white stone structures with small windows stand guard at the entrance of Topkapı Palace, which features a large archway and battlements. A red flag waves overhead, surrounded by verdant trees and under a clear azure sky. | (1) Two majestic gray stone towers with narrow openings dominate the entrance of Dolmabahçe Palace, which boasts a grand entrance arch and turrets. A yellow flag flutters above, encircled by lush greenery and beneath a bright blue sky.
(2) Three impressive beige stone edifices with tiny windows stand watch at the gateway of Topkapi Palace, which showcases a wide arch and crenelations. A green flag flaps in the breeze, amidst flourishing trees and under a clear sky.
(3) One towering white marble structure with large windows stands prominently at the entrance of Topkapı Palace, which features a colossal arch and fortified walls. A blue flag flutters above, complemented by blooming trees and set against an open sky. |

Figure 20: Examples of pseudo initial captions generated using the prompt in Figure 18, from MS COCO (top) and ShareGPT4V & DCI (bottom).

