# OpenReview forum: "One More Glance with Sharp Eyes: Rethinking Lightweight Captioning as a Practical Visual Specialist"
_ICLR.cc/2026/Conference — ICLR 2026 Conference Withdrawn Submission_

### Official Review · Reviewer_LKYY · 2025-10-24

**Soundness:** 2
**Presentation:** 3
**Contribution:** 2
**Rating:** 4
**Confidence:** 2

**Summary:**

This paper addresses the critical challenge of deploying large, computationally expensive Multimodal Language Models (MLLMs) for image captioning on resource-constrained edge devices. The authors begin by establishing a strong yet lightweight baseline model, or specialist, by replacing the LLaMA-7B language model in the LLaVA framework with OPT-125M, a model 56 times smaller. This lightweight specialist demonstrates surprisingly robust performance, achieving results on fact-based captioning tasks that are comparable to those of massive MLLMs. Despite this success, the lightweight model still produces intermittent errors. The authors diagnose the root cause as visual blindness, which they analyze as stemming from two specific failures: (1) a diffuse and inefficient attention mechanism that fails to focus on critical image regions during single-pass generation, and (2) coarse-grained, ambiguous visual representations from the CLIP encoder that lack sufficient detail for accurate captioning. To mitigate these issues, the paper introduces its core contribution: the Sharp-Eyed Refinement (SeR) framework. In the SeR framework, the model first generates a coarse initial caption. This caption, along with multi-layer visual features from the vision encoder, is then processed by the key DeepLens module. DeepLens leverages the initial text to guide the model's attention toward salient regions, enabling it to generate a more accurate and detailed final caption using richer, multi-layer visual features. Comprehensive experiments across captioning and long-range video QA tasks demonstrate that this framework outperforms existing lightweight methods and even large MLLMs. This is all achieved while using 93% fewer parameters and offering 97% faster inference than LLaVA-1.5-7B, proving the superiority of the proposed model.

**Strengths:**

- Achieving Comparable Performance with a Significantly Smaller Model: The paper's primary strength lies in achieving performance comparable to giant models with an extremely small, lightweight MLLM. The 450M-parameter lightweight specialist model achieves a CIDEr score of 129.6 on MS COCO, which is highly competitive with MLLMs more than 15 times its size.
- Insightful Root Cause Analysis: Instead of merely presenting a new architecture, the authors first identify the root causes of the single-pass model's failures. Through insightful analysis, including attention map visualizations and image reconstructions from visual features, they show that diffuse attention and coarse feature representations are the key bottlenecks, providing strong motivation for their proposed solution.
- Novelty of the Self-Refinement Framework: The Sharp-Eyed Refinement (SeR) framework itself represents a significant conceptual advance in multimodal learning. It successfully applies the concept of self-refinement to a multimodal context. This creates an effective feedback loop where the model's initial linguistic output directs its subsequent visual perception, moving from static, one-way generation to a more dynamic and iterative form of scene understanding. The framework's efficacy is clearly demonstrated through substantial performance improvements across multiple benchmarks.
- Comprehensive Validation and Practical Utility: Finally, the study is distinguished by its comprehensive empirical validation and clear demonstration of practical utility. The authors evaluate their model across various datasets and metrics and, crucially, test its performance on a downstream long-range video QA task. In this practical application, the 500M-parameter model achieves competitive accuracy with LLaVA-1.5-7B but completes the task in a fraction of the time. The successful deployment and testing on resource-constrained devices where the 7B model fails due to memory limitations also provide definitive proof of the solution's real-world applicability.

**Weaknesses:**

- Limited Standalone Practicality and Dependency on LLMs: The model's practicality as a standalone specialist is severely limited, highlighting its awkward strategic position. The authors' primary showcased application, long-range video QA, perfectly illustrates this dependency. This lightweight model's captions must be fed to a separate, massive 14B parameter LLM (Qwen2.5 14B) for the task to be completed. This pipeline undermines the entire premise of an efficient, on-device solution. For a user on a smartphone wanting to ask various visual questions, this model is insufficient on its own; it cannot answer general visual questions, a capability the paper itself defers as future work. A user would be far better served by a single, small generalist MLLM that can competently handle various tasks like VQA and OCR from the outset, rather than this dependent, captioning-only model.
- Lack of Quantitative Evidence for Root Cause Analysis: The authors diagnose the model's failure mode as visual blindness, citing (1) diffuse attention and (2) coarse visual representations as evidence. However, this claim relies solely on qualitative data, such as attention map visualizations (Figure 5) and image reconstruction results (Figure 6). To substantiate this argument, a quantitative evaluation is essential. For instance, the analysis would have been much more credible if the authors had measured how well the attention maps concentrate on actual object regions using metrics like bounding box Intersection over Union (IoU), or if they had evaluated the quality of the reconstructed images using methods like embedding similarity or MLLM-as-a-judge. Without quantitative data, the proposed diagnosis of the root cause lacks persuasive force.
- Latency Issues and Efficiency Limits of the SeR Framework: The SeR framework improves performance at the clear cost of increased latency. Due to its 2-pass architecture, it requires an additional forward pass through the language model, which increases inference time for image processing. In on-device applications where real-time interaction is critical (e.g., asking questions about objects viewed through a smartphone camera in real time), this latency can negatively impact the user experience. Furthermore, even with repeated applications of the refinement process, performance gains on the OPT-125M model quickly saturate and remain minimal. This shows a clear limit to the performance improvement gained in exchange for latency, suggesting that a single-pass model of a similar size might be a more practical choice if its average performance across multiple tasks is comparable, even if its captioning task performance is lower.

In conclusion, the previously detailed weaknesses—the model's fundamental dependency on large LLMs and the self-contradictory latency of its 2-pass architecture —place this 450M model in an untenable strategic position. It is stranded in an awkward middle ground between two more viable development paths. It lacks the versatility of the small generalist MLLMs (1-2B) that are increasingly viable for on-device use. Simultaneously, it fails to achieve the extreme efficiency and minimal 1-pass latency required for the very real-time applications it targets, a goal better suited for even smaller, hyper-efficient specialist models. By offering a 2-pass solution, the paper presents a methodology that is neither versatile enough to compete with generalists nor lean enough to be a true, fast specialist, making its practical utility highly questionable in a rapidly evolving field.

**Questions:**

- The root cause analysis of 'visual blindness' relies on qualitative visualizations (Figures 5 and 6). Could you provide quantitative validation, such as IoU for attention maps or quality metrics for image reconstructions?
- How do you justify the latency increase from the 2-pass SeR framework for your stated goal of real-time applications, especially given the minimal gains from iterative refinement?
- Fine-tuning relies on GPT-4o-mini to generate pseudo-initial captions. Did you experiment with using an open-source model?
- Beyond its specialized captioning ability, can the model demonstrate competent performance on general vision-language tasks such as classification or standard VQA benchmarks (e.g., VQAv2, GQA, OCR)?
- Your paper's primary comparisons, LLaVA-1.5 and Qwen-VL, are based on models from 2023-2024. However, the field of small generalist MLLMs is advancing rapidly. How does your model's performance and latency compare against current sota models like Qwen-VL 2.5 or InternVL 3.5? (before 2025.09)

---

### Official Review · Reviewer_kQDy · 2025-10-28

**Soundness:** 3
**Presentation:** 3
**Contribution:** 2
**Rating:** 4
**Confidence:** 5

**Summary:**

This paper introduces Sharp-Eyed Refinement, an image captioning framework that leverages the model's learned self-refinement capability. The authors first demonstrate that image captioning tasks do not require the high reasoning ability of LLMs by comparing the OPT-125M-based model with other models using much larger LMs. Motivated by human behaviors and attention analysis, the authors propose a two-step caption generation method. The proposed method 1) generates a caption; 2) updates the initial caption using the DeepLens module. The module takes an image-caption pair as its input and produces the input tokens for the following LM. The authors train DeepLens on perturbed-corrected caption pairs generated by GPT-4o-mini. By slightly changing key components in images like entities, attributes, and relations in purpose, DeepLens is guided to focus more on important features presented in images.

**Strengths:**

1. This paper is clear and well-organized overall.
2. The problem addressed in the paper is interesting and timely.

**Weaknesses:**

1. Note 2: The issue of visual attention patterns has already been studied in several existing studies [1,2].
2. Note 3: The issue of CLIP embeddings in terms of image generation and understanding has already been studied in many studies [3,4].
3. Is this approach scalable? If you use bigger models, can you still see the improvements?

[1] Liu et al., "Paying more attention to image: A training-free method for alleviating hallucination in lvlms" ECCV 2024
[2] Jung et al., "Visual Attention Never Fades: Selective Progressive Attention ReCalibration for Detailed Image Captioning in Multimodal Large Language Models" ICML 2025
[3] Yuksekgonul et al., "When and why vision-language models behave like bags-of-words, and what to do about it?" ICLR 2023
[4] Lin et al., "Evaluating Text-to-Visual Generation with Image-to-Text Generation" ECCV 2024

**Questions:**

Is this approach scalable? If you use bigger models, can you still see the improvements?

---

### Official Review · Reviewer_Avmn · 2025-10-31

**Soundness:** 3
**Presentation:** 3
**Contribution:** 3
**Rating:** 6
**Confidence:** 2

**Summary:**

The paper reframes image captioning as a lightweight “visual specialist” problem for on-device deployment. Concretely, the authors build captioners around a 125M-parameter LM and introduce Sharp-Eyed Refinement: a two-stage “coarse-to-refine” procedure that first produces a rough caption and then revisits salient regions to sharpen details. A module called DeepLens is used to “look again” and strengthen visual grounding during the refinement stage. The approach matches or approaches MLLM performance on detailed captioning and also shows gains on long-range video QA, suggesting practical potential for on-device assistants.

**Strengths:**

Well-motivated objective: Targeting on-device captioning with a compact LM addresses a real deployment gap.
Simple method: SeR (coarse to refine) and DeepLens (re-examining informative regions) align with how humans revisit images to recover missed details.

**Weaknesses:**

1. It’s not stated whether MLLM baselines were fine-tuned under the same data/prompts or used zero-shot.
2. The effectiveness of each SeR/DeepLens component and iteration count are not ablated.
3. The paper doesn’t show device-level latency/memory results on real hardwares.

**Questions:**

see weakness

---

### Official Review · Reviewer_KgkF · 2025-11-01

**Soundness:** 2
**Presentation:** 2
**Contribution:** 2
**Rating:** 2
**Confidence:** 4

**Summary:**

This paper addresses MLLMs’ high deployment cost for image captioning on edge devices. It builds a small parameter lightweight model, which matches MLLMs’ performance on single-sentence/detailed captioning tasks. To fix its inaccuracies from diffuse attention and coarse CLIP features, it proposes the SeR framework with DeepLens. Two-stage training enables refinement.

**Strengths:**

1. The paper improves the lightweight baseline on image captioning tasks. The improvements seem consistent with the extra modules and parameters.
2. A notable strength of this manuscript lies in the authors’ detailed and well-substantiated elaboration of the research motivation, which clearly articulates the rationale and necessity of the present study.

**Weaknesses:**

1. The paper introduces many new concepts, like SeR and DeepLens, but fails to introduce the clear technical implementation. The presentation of this paper introduces many Hypotheses and Notes, but they lack empirical support and have no theoretical guarantee or reference.

2. The analysis part does not introduce new ideas or novel views towards the problem, and the experiments do not verify the hypotheses. For example, the architecture of MAE is not built for image reconstruction, but for pre-training. Therefore, the architecture itself cannot guarantee good quality.

3. The architecture and design introduced in the appendix are well-studied: multi-layer fusion or Q-former-style fusion has been used for years, which makes DeepLens lack novelty.

**Questions:**

1. Figure 4 is not referenced in the paragraph. If OPT is so poor, why use it? How does SeR work on stronger backbones?
2. The paper adopts a very small LLM (OPT-125M) but uses a much larger ViT. Why not use a small ViT with a larger LLM?
3. Can the method of this paper apply to fields besides captioning? None of the designs are restricted to the field of captioning.

---

### Note · Authors · 2025-11-14

**Comment:**

Thank you to the reviewers and ACs for the time and effort you dedicated to our paper. We will carefully incorporate your feedback and plan to submit an improved version to a future venue. Thank you again for the insightful and valuable comments.

**Withdrawal Confirmation:**

I have read and agree with the venue's withdrawal policy on behalf of myself and my co-authors.